



# Multi-level data assimilation for simplified ocean models

Florian Beiser[1,2,], Håvard Heitlo Holm[1], Kjetil Olsen Lye[1], and Jo Eidsvik[2]

[1]*Mathematics and Cybernetics, SINTEF Digital, Oslo, Norway*
[2]*Department of Mathematical Sciences, NTNU, Trondheim, Norway*

**Correspondence:** Florian Beiser (florian.beiser@sintef.no)

**Abstract.** Multi-level Monte Carlo methods have established as a tool in uncertainty quantification for decreasing the computational costs while maintaining the same statistical accuracy as in single-level Monte Carlo. Lately, there have also been theoretical efforts to use similar ideas to facilitate multi-level data assimilation. By applying a multi-level ensemble Kalman filter for assimilating sparse observations of ocean currents into a simplified ocean model based on the shallow-water equations,

we study the practical challenges of applying these method to more complex problems. We present numerical results from a realistic test case where small-scale perturbations lead to chaotic behaviour, and in this context we conduct state estimation and drift trajectories forecasting using multi-level ensembles. This represents a new step on the path of making multi-level data assimilation feasible for real-world oceanographic applications.

**Short Summary**

Efficient search-and-rescue at sea is supported by rapid forecasts of drift trajectories based on a dynamical model and data assimilation of in situ observations. The multi-level approach can accelerate computations by running models on various resolutions. This work explores the practicality of multi-level methods for realistic data assimilation applications by considering a challenging shallow-water case for the ocean dynamics.

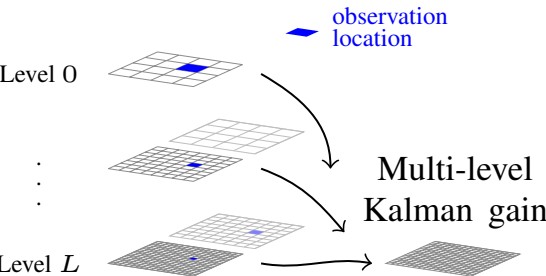

**Figure 0.** Key figure for graphical abstract



# 1 Introduction

The multi-level Monte Carlo (MLMC) method leverages ensemble simulations of varying accuracy to approximate statistical quantities (Giles, 2008). By harnessing the cost-effectiveness of low-fidelity simulations within the ensemble, MLMC estimators achieve the same theoretically statistical accuracy as classical single-level approaches but at significantly reduced computational expenses. In spatio-temporal applications, the different levels often correspond to solving the same problem on varied grid resolutions, where there is a decrease in computational costs with lower resolution (Mishra and Schwab, 2012).

This approach has become an established tool in the realm of uncertainty quantification, primarily due to its computational efficiency (Giles, 2015). Recently, the adaptation of multi-level methodologies has extended into the field of data assimilation, aimed at updating ensembles of numerical models with partial and noisy observations of the system. While related work on multi-level data assimilation (MLDA) has largely concentrated on theoretical analyses of potential gains (Hoel et al., 2016; Beskos et al., 2017; Jasra et al., 2020; Chernov et al., 2021), our study distinguish itself by applying these methods to a practical
scenario of real-world relevance.

Our motivation to exploit MLDA arises from search-and-rescue (SAR) operations at sea, where efficient prediction of drift trajectories with associated uncertainties is of utmost importance. Drift trajectory models used operationally (e.g., Dagestad et al. (2018)) typically compute the trajectories by using already available forecasts of the ocean currents generated by generalised circulation model (e.g., Shchepetkin and McWilliams (2005)). Circulation models are typically expensive to run on
the computer, meaning that the available input fields for the trajectory model are either from a deterministic forecast, or a small ensemble at best. To complement this approach, one has suggested to run larger ensembles of simplified ocean models to investigate the uncertainty of the ocean currents, facilitate data assimilation, and consequently quantify the uncertainty in the drift trajectories (Røed, 2012). By obtaining in situ observations during the SAR operation, such an ensemble can furthermore be updated efficiently through data assimilation (Holm et al., 2020a, b).

Within this context, we aim to apply and investigate the properties of the recently proposed MLDA methods within an existing GPU-accelerated framework for running ensembles of simplified ocean models (Brodtkorb and Holm, 2021). The simulation framework builds on finite-volume schemes, and we will therefore formulate the MLDA in that setting. The driving questions for carrying out this work revolve around assessing the potential gain in computational efficiency as indicated by the existing theoretical analysis and the practicability of the approach. More specifically, the key research questions we address
are:

    i) How can one extend an existing simulation framework designed for single-level ensembles to run multi-level ensembles? And which performance gain can MLDA deliver when we do not have perfect run time scaling?

    ii) Often in MLMC, one optimises the ensemble sizes with respect to a single statistical quantity of interest, but in data assimilation one is interested in several quantities of interests, among others state mean, variance, and correlations with
45         observations. How can one still configure a multi-level ensemble for MLDA?





iii) How does the configuration of multi-level ensemble influence the quality of the data assimilation? And how does these MLDA results compare to equivalent data assimilation methods for traditional single-level ensembles?

iv) How can a multi-level ensemble be used for drift trajectory forecasting?

## 1.1 Related work

MLMC is an efficient approximation method for quantities of interest that involve high-dimensional integrals, and it was originally introduced for numerical integration (Heinrich, 2001). For uncertainty quantification, the method was first presented for the estimation of expected values (Giles, 2008). The multi-level estimator was then constructed by using approximations of increasing accuracy, by choosing ensembles on the different levels and employing a telescopic sum (Giles, 2015). For this concept, Schaden and Ullmann (2020) first derived ways to construct the best linear unbiased estimators in the scalar case with

fixed budget, before Destouches et al. (2023) did the same in the multi-dimensional case. For obtaining numerical solutions to partial differential equations (PDEs), the hierarchy of levels was constructed by solving the problem on increasingly fine grids, and using the MLMC on the results (Dodwell et al., 2015). The case of hyperbolic conservation laws and finite volume methods have been separately analysed for the application of MLMC by Mishra and Schwab (2012) and Fjordholm et al. (2020), including the particular model of the shallow-water equations (Mishra et al., 2012a).

Ensemble-based data assimilation is a set of techniques that incorporate observations into a probabilistic forecast generated by running an ensemble of numerical models (Evensen et al., 2022). The nature of the methods are commonly split into ensemble Kalman filters (EnKF), particle filters (PF), and ensemble variational methods (EnVar) and an overview with emphasis on geophysical problems can be found in Carrassi et al. (2018) and Vetra-Carvalho et al. (2018). Adaptions to the characteristics of the model or the data exist, and include approaches for drifter observations (Sun and Penny, 2019).

PFs are non-linear filters that in their purest form evaluate Bayes' rule directly by adjusting the relative importance or weights of each ensemble member according to its resemblance to the observations (Van Leeuwen et al., 2019). In the multi-level version of PFs, the weights are updated first on the coarsest level as in the single-level PF, and then on increasingly finer coupled levels (Jasra et al., 2017; Latz et al., 2018). In contrast, variational methods solve a numerical optimisation problem that seeks to minimise the discrepancy between the numerical model and the observations with respect to the observation uncertainties,

and the difference between the prior state and the posterior with respect to the so-called background error covariances. EnVar seeks to estimate the background error covariance using an ensemble, and multi-level ensembles can been used for this purpose (Bierig and Chernov, 2014; Mycek and De Lozzo, 2019).

Our interest lies in EnKFs, which are derived from the linear Gaussian model, but still proves highly useful in many non-linear applications and are therefore popular in practice (Evensen, 2006). The multi-level EnKF (MLEnKF) was first introduced

for temporal processes by Hoel et al. (2016), where the multi-level estimator is applied to the Kalman Gain. This result was extended to spatio-temporal processes by Chernov et al. (2021), and similar approaches were developed in history matching where the multi-level ensembles are complemented with hybrid filters (Fossum et al., 2020; Nezhadali et al., 2020).





## 1.2 Contribution and outline

In this work, we discuss and assess the applicability of multi-level data assimilation techniques to practical ocean forecasting
applications. We do this through using simplified ocean models represented by rotating shallow-water equations, assimilating
observation from a particular challenging test case, before using the posterior multi-level ensemble to forecast drift trajectories.
In this regard, this work brings MLDA one step further from proof-of-concept towards application in operational models.

First, we revise the MLEnKF introduced by Chernov et al. (2021) to a finite-volume framework and incorporate localisation
seamlessly into the multi-level structure, enabling robust data assimilation. Subsequently, we direct the attention on a set
of emerging practical challenges and suggest pragmatic solutions. Among others, we work with the assumptions of both
theoretical and practical computational work and explain how it influences the performance of MLDA. Furthermore, we use an
existing GPU-accelerated framework for shallow-water simulations, which is optimised for fine spatial resolution, and set up a
data assimilation case with opposing jets. We assimilation sparse observations using the MLEnKF. For this system, we present
that the numerical results show the same state estimation quality as a single-level EnKF of corresponding theoretical error, but
we obtain a speed-up of up to a factor 2. Lastly, we demonstrate further practical use of multi-level ensembles in Sect. 5 by
showing how to use the posterior ensemble for drift trajectory modelling and forecasting.

The rest of the paper is structured as follows. In Sect. 2, we introduce the general data assimilation problem, repeat the basis
for multi-level statistics, and describe the MLEnKF in the context of numerical models based on finite-volume methods. In
Sect. 3, we elaborate on practicalities in the implementation of MLDA. In Sect. 4, we compare numerical results for single-
level EnKF and MLEnKF. Sect. 5 is dedicated to forecasting of drift trajectories. In Sect. 6, we finish with some concluding
remarks.

## 2   Multi-level data assimilation

We start by introducing the data assimilation problem as it appears in ocean forecasting, and then recap the MLMC approach
in this context to outline the suggested MLEnKF in a form that is applicable to models based on finite-volume schemes. This
lays the ground for the specific discussion of practical issues and implementation details in the remainder of the article.

### 2.1   The data assimilation problem

Data assimilation is a central methodology that integrates observational data into a numerical representation of the same system.
We consider a state variable $\boldsymbol{x}(t) \in \mathbb{R}^{n_x}$, which is evolved forward in time by

$$\boldsymbol{x}(t + \Delta t) = \mathcal{M}(\boldsymbol{x}(t)) + \delta\boldsymbol{x}, \tag{1}$$

where $\mathcal{M}$ is the model operator and $\delta\boldsymbol{x}$ is a random model error term that represents missing or unresolved physics in the
model. In our application, $\mathcal{M}$ will be a simplified ocean model solved numerically by a finite-volume method on a regular
Cartesian grid. The state $\boldsymbol{x}(t)$ will therefore represent grid cell averages of physical variables for all grid cells in the domain





at time $t$. This means that the size of the state vector $n_x$ will be the product of the number of grid cells in the domain and the number of physical variables included in the model.

The computational cost of simulating the forward model in (1) depends on the numerical model itself and on the chosen discretisation. In general, coarser resolutions are significantly faster to run than finer ones. For a two-dimensional grid with cell edges of size $\Delta x$, the computational effort to solve (1) numerically is

$$\mathcal{O}(\Delta x^{-3}), \tag{2}$$

see, e.g., Mishra et al. (2012a). Here, one magnitude comes from the fact that the numerical solver has to take time steps

proportional to $\Delta x$. This means that doubling $\Delta x$ or equivalently four-dividing the number of grid cells (in two dimensions) reduces the computational cost eight times.

    In sequential data assimilation, the state is evolved by finite model time steps and we are interested in the incorporation of data at the isolated model time steps where all quantities are at the same point in time. Hence, unless otherwise stated, we omit notation of time in the state vector for better readability, and we set $\boldsymbol{x} = \boldsymbol{x}(t)$.

We consider incomplete and noisy observations $\boldsymbol{y} \in \mathbb{R}^{n_y}$ of the true state of the simulated system denoted $\boldsymbol{x}_{\text{true}}$. The model for the observation is

$$\boldsymbol{y} = \mathbf{H}\boldsymbol{x}_{\text{true}} + \boldsymbol{\varepsilon}, \tag{3}$$

where $\mathbf{H}$ is the observation operator and $\boldsymbol{\varepsilon} \sim \mathcal{N}(0, \mathbf{R})$ is the observation noise, which we assume to be Gaussian with a diagonal covariance matrix $\mathbf{R}$. Furthermore, we restrict ourselves to linear observation operators mapping from the state space to the

observational space, meaning $\mathbf{H} \in \mathbb{R}^{n_y \times n_x}$.

    With the forward model and the observations in place, the data assimilation problem consists of updating the prior (empirical) probability density function $p(\boldsymbol{x})$ of the state $\boldsymbol{x}$ with the observations $\boldsymbol{y}$ to obtain the conditional posterior probability density function $p(\boldsymbol{x}|\boldsymbol{y})$. This is done using Bayes' theorem. In the rest of this paper, we consider sequential data assimilation in terms of filtering, meaning that the model in (1) is only used to evolve the state between observations times.

**2.2   Single-level EnKF**

To represent the probability distributions for the state, we use ensembles $(\boldsymbol{x}_e)_{e=1}^{N}$ which facilitate Monte Carlo estimation to infer statistical quantities of interest. In the classical case, we use the notation

$$\overline{\boldsymbol{x}} = \frac{1}{N} \sum_{e=1}^{N} \boldsymbol{x}_e \tag{4a}$$

and

$$\tilde{\boldsymbol{x}}_e = \boldsymbol{x}_e - \overline{\boldsymbol{x}} \tag{4b}$$

for the ensemble average and anomalies, respectively. Then, the unbiased single-level estimators for the mean and covariance are

$$\boldsymbol{\mu}^{\text{SL}}[\boldsymbol{x}] = \overline{\boldsymbol{x}} \tag{5a}$$





and

$$\boldsymbol{\Sigma}^{\mathrm{SL}}[\boldsymbol{x},\boldsymbol{x}] = \frac{1}{N-1}\sum_{e=1}^{N}\tilde{\boldsymbol{x}}_e(\tilde{\boldsymbol{x}}_e)^{\top}. \tag{5b}$$

Here, the statistical quality of the estimator depends on the number of ensemble members, the quantity of interest as well as the grid resolution. In terms of computer resources, there is a trade off between the grid resolution and the ensemble size, and it is accepted that the same ensemble approximates the mean and the covariance with different accuracy. Similarly, the mean and the anomalies in the modelled ensemble observations without noise are denoted by

$$\overline{\boldsymbol{y}} = \frac{1}{N}\sum_{e=1}^{N}\mathbf{H}\boldsymbol{x}_e \tag{6a}$$

and

$$\tilde{\boldsymbol{y}}_e = \mathbf{H}\boldsymbol{x}_e - \overline{\boldsymbol{y}}, \tag{6b}$$

respectively.

The EnKF is a popular technique for assimilating observation data into an ensemble. In this approach, covariances between the state and data variables are computed from the ensembles, and in doing so, one also forms a Monte Carlo approximation of the Kalman gain $\mathbf{K}$. The state vectors in the ensemble are then linearly updated, see, e.g., Evensen (2006) and Carrassi et al. (2018). In sequential data assimilation routines, this procedure is then repeated at all data gathering times.

For the single-level EnKF, the Kalman gain can be approximated by

$$\mathbf{K}^{\mathrm{SL}} = \boldsymbol{\Sigma}_{XY}^{\mathrm{SL}}(\boldsymbol{\Sigma}_{YY}^{\mathrm{SL}})^{-1}, \tag{7}$$

with

$$\boldsymbol{\Sigma}_{XY}^{\mathrm{SL}} = \frac{1}{N-1}\sum_{e=0}^{N}\tilde{\boldsymbol{x}}_e(\mathbf{H}\tilde{\boldsymbol{x}}_e)^{\top} \tag{8a}$$

and

$$\boldsymbol{\Sigma}_{YY}^{\mathrm{SL}} = \mathbf{H}\boldsymbol{\Sigma}_{XY}^{\mathrm{SL}} + \mathbf{R}. \tag{8b}$$

Here, $\boldsymbol{\Sigma}_{XY}^{\mathrm{SL}}$ estimates the cross-covariance between the state and the corresponding predicted observations, while $\boldsymbol{\Sigma}_{YY}^{\mathrm{SL}}$ is the covariance between the observations. Note that the construction of a large state covariance matrix is circumvented in (8). Then, the ensemble is updated by

$$\boldsymbol{x}_e^a = \boldsymbol{x}_e + \mathbf{K}^{\mathrm{SL}}(\boldsymbol{y} + \boldsymbol{\varepsilon}_e - \mathbf{H}\boldsymbol{x}_e), \quad e = 1,\ldots,N, \tag{9}$$

where the superscript $a$ marks the posterior (often called the analysis) states, and $\boldsymbol{\varepsilon}_e$ are independent realisations of the observation noise. Various algebraically equivalent forms of the Kalman update exist and in practice the cross-covariance or the Kalman gain matrix is often tapered away from observation indices to avoid spurious correlations that are theoretically irrelevant but naturally appear because of Monte Carlo variability.



### 2.3 Multi-level Monte Carlo

We recall the basics of MLMC for the sake of completeness and for introducing consistent notation.

#### 2.3.1 Up- and downscaling

The multi-level approach relies on the definition of a hierarchy of grids. Starting with the original grid of interest that has the finest resolution, computationally less expensive grids are defined. The steps in this hierarchy of discretisations are called levels and denoted with superscripts $0, \dots, L$ where $L$ is the original finest level. The corresponding states are represented by $\boldsymbol{x}^l \in \mathbb{R}^{n_x^l}$ for $l = 0, \dots, L$ with $n_x^0 < \dots < n_x^L$ as vectors of different dimensions, where $n_x^L = n_x$ for consistency with the single-level notation.

For the projections between the different levels, it becomes computationally convenient to nest grids. There are several possible techniques for nesting a finite-dimensional vector, but it is helpful to take into consideration what the state variables actually represent (Farmer, 2002). For example for point values linear interpolation may be adequate. Given a two-dimensional grid for a finite-volume scheme, where the state variables represent average values for the grid cells, coarser grids can be built by iteratively merging neighbouring cells from the next finer grid. Note that we restrict ourselves here to a coarsening

factor of 2 as it is computationally most convenient, and then nested grids are generated by merging two times two cells each. For nested meshes and in particular for the one-cell to/from four-cells scheme, simple repeating/averaging provides feasible up/downscaling which is conservative in accordance with the finite-volume framework and also computationally efficient. This enables us to execute algebraic calculations for states on different levels, but for improved readability, the up- and downscaling is never explicitly denoted.

#### 2.3.2 Multi-level estimators

A multi-level ensemble consists of coarse members and pairs of states members at subsequent levels:

$$\{\boldsymbol{x}_e^0\}_{e=1}^{N^0}, \quad \{\boldsymbol{x}_e^{l+}, \quad \boldsymbol{x}_e^{l-}\}_{e=1}^{N^l}, \quad l = 1, \dots, L. \tag{10}$$

Here, the partners $\boldsymbol{x}_e^{l+}$ and $\boldsymbol{x}_e^{l-}$ in a pair share the same realisation, but $\boldsymbol{x}^{l+}$ is defined on level $l$ and $\boldsymbol{x}^{l-}$ on level $l-1$. Note that, e.g., $\boldsymbol{x}^{2-}$ and $\boldsymbol{x}^{1+}$ are defined on the same level and hence, the notation with $+$ and $-$ in the superscript is used to

distinguish them. For simplicity, we let $\boldsymbol{x}^L = \boldsymbol{x}^{L+}$ as these are the only ensemble members defined on the finest grid.

    According to the level dimensions, a hierarchy of model operators $\mathcal{M}^l$, $l = 0, \dots, L$ evolves states on the different grids in time. This means, for example, that $\boldsymbol{x}^{l+}$ is evolved by $\mathcal{M}^l$ and $\boldsymbol{x}^{l-}$ by $\mathcal{M}^{l-1}$. In the case of consistent model equations, the different model operators $\mathcal{M}^l$ represent the same numerical solver with different underlying grids according to the level definition. Similarly, the model perturbations $\delta \boldsymbol{x}^l$ are projections from the original noise on the finest resolution.

A multi-level estimator utilises a telescoping sum over the levels for the estimation of quantities of interest, and this construction makes it different from the typical empirical approximation of classical Monte Carlo estimators. For the expected




value, we have the following telescoping sum property

$$\mathbb{E}\left(\boldsymbol{x}^L\right) = \sum_{l=1}^{L} \mathbb{E}\left(\boldsymbol{x}^{l+} - \boldsymbol{x}^{l-}\right) + \mathbb{E}(\boldsymbol{x}^0), \tag{11}$$

and in general, for a function $g : \mathbb{R}^{n_x} \to \mathbb{R}^{n_g}$ for some $n_g \in \mathbb{N}$, we have

$$\mathbb{E}\left(g(\boldsymbol{x}^L)\right) = \sum_{l=1}^{L} \mathbb{E}\left(g(\boldsymbol{x}^{l+}) - g(\boldsymbol{x}^{l-})\right) + \mathbb{E}(g(\boldsymbol{x}^0)). \tag{12}$$

The above motivates the following general approximation

$$\mathbb{E}\left(g(\boldsymbol{x}^L)\right) \approx \sum_{l=1}^{L} \left[ \frac{1}{N^l} \sum_{e=1}^{N^l} \left(g(\boldsymbol{x}^{l+}) - g(\boldsymbol{x}^{l-})\right) \right] + \frac{1}{N^0} \sum_{e=1}^{N^0} g(\boldsymbol{x}^0) \tag{13}$$

using a multi-level ensemble of samples. Following the above expression, one can define multi-level estimators for the mean and covariance

$$\boldsymbol{\mu}^{\mathrm{ML}}[\boldsymbol{x}^L] = \overline{\boldsymbol{x}}^0 + \sum_{l=1}^{L} (\overline{\boldsymbol{x}}^{l+} - \overline{\boldsymbol{x}}^{l-}) \tag{14a}$$

and

$$\boldsymbol{\Sigma}^{\mathrm{ML}}[\boldsymbol{x}^L, \boldsymbol{x}^L] = \boldsymbol{\Sigma}^0 + \sum_{l=0}^{L} (\boldsymbol{\Sigma}^{l+} - \boldsymbol{\Sigma}^{l-}), \tag{14b}$$

where $\overline{\boldsymbol{x}}^{l+}$ and $\boldsymbol{\Sigma}^{l+}$ are the single-level mean and covariance estimators on level $l$ estimated by the ensemble $(\boldsymbol{x}_e^{l+})_{e=1}^{N_e}$ only. The analogous notation is used for the $(\boldsymbol{x}_e^{l-})_{e=1}^{N_e}$ ensemble. By the use of linearity, the covariance at the finest level becomes

$$\boldsymbol{\Sigma}^{\mathrm{ML}}[\boldsymbol{x}^L, \boldsymbol{x}^L] = \frac{1}{N^0 - 1} \sum_{e=0}^{N^0} \tilde{\boldsymbol{x}}_e^0 (\tilde{\boldsymbol{x}}_e^0)^\top + \sum_{l=1}^{L} \frac{1}{N^l - 1} \sum_{e=1}^{N^l} \left( \tilde{\boldsymbol{x}}_e^{l+} (\tilde{\boldsymbol{x}}_e^{l+})^\top - \tilde{\boldsymbol{x}}_e^{l-} (\tilde{\boldsymbol{x}}_e^{l-})^\top \right), \tag{15}$$

where the differences are taken between the partners on every level directly.

### 2.3.3 Theoretical results

We now fix a function $g$ which is given by the statistic one wants to compute. Under certain assumptions, a multi-level estimator for $\mathbb{E}[g(\boldsymbol{x})]$ yields a computational speed-up for estimating the quantity of interest while keeping a fixed statistical accuracy $\tau^2$. Alternatively, from the reverse perspective, the estimator yields improved statistical accuracy while keeping the computational work fixed (Giles, 2015; Lye, 2020).

The main idea is that if the numerical discretisation convergences in the sense that $\|g(\boldsymbol{x}^l) - g(\boldsymbol{x}^{l-1})\|_2 \to 0$ almost surely as $l \to \infty$, then the variance of the details $\mathbb{V}\mathrm{ar}[g(\boldsymbol{x}^{l+}) - g(\boldsymbol{x}^{l-})] = \mathcal{O}(\mathbb{E}(\|g(\boldsymbol{x}^l) - g(\boldsymbol{x}^{l-1})\|_2^2))$ will be small for large $l$. In other words, the sampling error on the finer levels is small, meaning we need fewer samples on the finer, more expensive levels. On the coarser levels we will need more samples, but the coarser levels are cheaper to compute due to (2).





Let $c^l$ denote the cost of generating a sample $\boldsymbol{x}^l$ at level $l$ and let $C^l$ be the cost of $\boldsymbol{x}^{l+} - \boldsymbol{x}^{l-}$ which actually involves two simulations, where $C^0 = c^0$ by definition. The cost is determined by the computational work it takes to solve the PDE numerically with some software on some hardware. The theoretical costs can be specified using (2) or must be found in practice by measuring the run time of the simulation for different numerical grid sizes. Analogously, let $\|\cdot\|_2$ be the $L^2$-norm over the domain, then let

$$v^l = \|\mathbb{V}\mathrm{ar}[g(\boldsymbol{x}^l)]\|_2 \tag{16a}$$

and

$$V^l = \|\mathbb{V}\mathrm{ar}[g(\boldsymbol{x}^{l+}) - g(\boldsymbol{x}^{l-})]\|_2, \tag{16b}$$

where we set $V^0 = v^0$. These variances can be estimated beforehand using experiments or theoretical considerations of convergence in the setting of the numerical discretisation used.

Based on such analysis, one can efficiently allocate computational resources, where by rule-of-thumb a speed-up is achieved if

$$V^l \ll v^0 \tag{17a}$$

and

$$C^0 \ll \cdots \ll C^L. \tag{17b}$$

For the best speed up, we need a strong correlation between the levels and a strong computational scaling for the different resolutions. Given a fixed number of levels $L$ and corresponding variances $\{V_l\}_{l=0}^L$ and computational costs $\{C_l\}_{l=0}^L$, one can derive optimal sample numbers and a theoretical speed-up against single-level Monte Carlo. We will do so in Sect. 3.1.3.

### 2.4 Multi-level ensemble Kalman filter

The application of the multi-level estimator for the Kalman gain approximation in spatio-temporal problems is originally suggested in Chernov et al. (2021). Our work builds on this and revises the method for hierarchically gridded data and explains the construction of the matrix operators. Note that the Kalman gain $\mathbf{K}^{\mathrm{ML}}$ is a matrix $\mathbb{R}^{n_x \times n_y}$ using the finest resolution. It can be constructed by the multiplication of the state-observation cross-covariance and the inverse of the observation covariance from (7) and we hence draw the attention to these matrices. Using the multi-level estimator for $\boldsymbol{\Sigma}_{XY}$, we arrive at

$$\boldsymbol{\Sigma}_{XY}^{\mathrm{ML}} = \frac{1}{N^0 - 1} \sum_{e=0}^{N^0} \tilde{\boldsymbol{x}}_e^0 (\tilde{\boldsymbol{y}}_e^0)^\top + \sum_{l=1}^{L} \frac{1}{N^l - 1} \sum_{e=1}^{N^l} \left( \tilde{\boldsymbol{x}}_e^{l+} (\tilde{\boldsymbol{y}}_e^{l+})^\top - \tilde{\boldsymbol{x}}_e^{l-} (\tilde{\boldsymbol{y}}_e^{l-})^\top \right), \tag{18}$$

where differences between partner levels are taken on the finer resolution and contributions of the different levels are upscaled in the end. The observation covariance matrix can be extracted from $\boldsymbol{\Sigma}_{XY}^{ML}$ as before in (8) or calculated independently from (14).





For updating the multi-level ensemble, the Kalman gain is downscaled onto the respective level. In the MLEnKF, the analysis
state then becomes for the coarsest level

$$\boldsymbol{x}_e^{0,a} = \boldsymbol{x}_e^0 + \mathbf{K}^{\mathrm{ML}}(\boldsymbol{y} + \boldsymbol{\varepsilon}_e^0 - \mathbf{H}\boldsymbol{x}_e^0), \quad e = 1,\ldots,N^0 \tag{19a}$$

and for the pairs on the higher level it is

$$\boldsymbol{x}_e^{l+,a} = \boldsymbol{x}_e^{l+} + \mathbf{K}^{\mathrm{ML}}(\boldsymbol{y} + \boldsymbol{\varepsilon}_e^l - \mathbf{H}\boldsymbol{x}_e^{l+}), \qquad \boldsymbol{x}_e^{l-,a} = \boldsymbol{x}_e^{l-} + \mathbf{K}^{\mathrm{ML}}(\boldsymbol{y} + \boldsymbol{\varepsilon}_e^l - \mathbf{H}\boldsymbol{x}_e^{l-}), \quad e = 1,\ldots,N^l. \tag{19b}$$

Here, the observation noise $\boldsymbol{\varepsilon}_e^l$ are independent realisations for all ensemble members on all levels, where pairs on higher
levels again share the same realisation.

In the estimation, we upscale recursively to the next level. This means that for the computation of the difference between
two levels $\boldsymbol{x}^{l+} - \boldsymbol{x}^{l-}$, we upscale $\boldsymbol{x}^{l-}$ to have the same resolution as $\boldsymbol{x}^{l+}$. Eventually, we end up with an estimator on the
finest level. In the ensemble update step, the spatial dimension of the Kalman gain is then downscaled to the respective level,
the observation dimension stays unmodified all the time. Moreover, simple algebraic transformations ensure that the order
of scaling and evaluation of statistical quantities does not matter. For example, the covariance matrix can be formed on a
coarse grid and then upscaled to a finer grid by applying the projection operator from one side and its transposed from the
other. Alternatively, the states can first be upscaled to the finer grid before the covariance matrix is assembled. Both of these
procedures give the same result.

Figure 1 shows a schematic illustration of conditioning with multi-level ensembles. Simulations of state variables and syn-
thetic data (left) are generated on the coarsest level and at partner levels. All these are used to fit the Kalman gain matrix which
is enables linear updating of the ensembles at various resolution levels with the observations (right).

With this implementation of the MLEnKF, we stay close to the formulation of Chernov et al. (2021). But even though we
employ a analogous estimation and updating scheme as Chernov et al. (2021), their theoretical assumptions no longer hold
in our set-up and it is further unlikely for realistic applications to have a mean-field reference. Nonetheless, we strive for a
computational speed-up through the MLEnKF, but we use more practical ways to assess the MLDA quality. The estimation
over nested grids also shares similarities with Fossum et al. (2020), but in contrast to them we avoid the hybridisation with
Bayesian averaging and we keep the higher levels coupled.

## 3   Practical considerations

Until now we have discussed MLDA and its potential using MLEnKF on a theoretical level. In the following, we discuss some
of the challenges we face and the choices that we have to make when applying MLEnKF to practical applications.

### 3.1   Multi-level ensemble sizes

The optimal configuration of the multi-level ensemble depends on the reduction of computational costs towards coarser grids,
but also on the quantity of interest that we want to estimated. In sequential data assimilation cycles, there are actually multiple





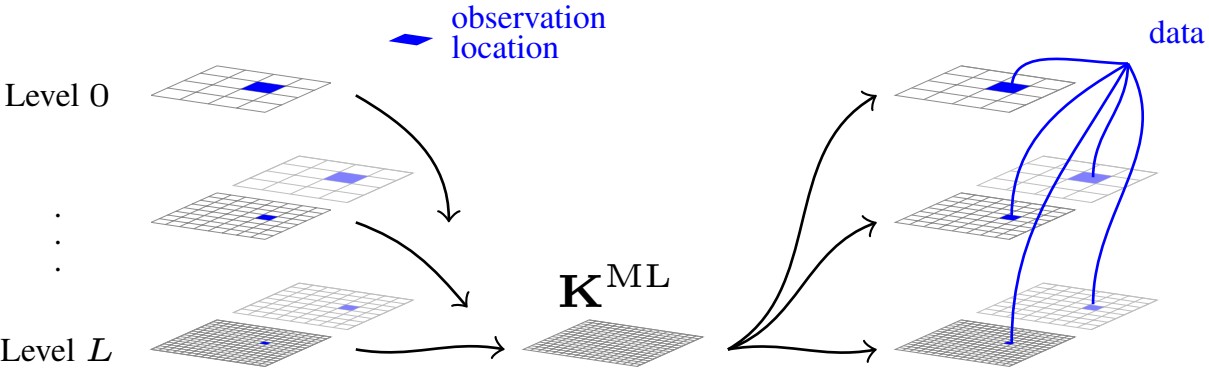

**Figure 1.** In multi-level conditioning with the MLEnKF, the empirical Kalman gain is assembled by a telescopic sum from the multi-level prior ensemble. Here, we use single observation location for illustration purposes. The Kalman gain is represented on the finest resolution and then projected to each level for updating. The update combines the prior ensemble with the data weighted by the Kalman gain for each ensemble member.

quantities of interest at different steps in time and at different locations. High-quality estimation of multi-level ensemble mean and standard deviation are important to make robust statements regarding the forecast. This again relies on the estimation of the Kalman gain which must be constructed at every update time from the covariance estimation. Assessment scores may also be used to evaluate the quality of the ensemble-based representation, and these tasks require various kinds of functions.

For single-level ensembles, the decision for an ensemble size is straightforward and is typically based on how many samples one can afford within the computational budget. In the multi-level case, the allocation of the computational budget is more involved. Often the assumption of exponentially growing ensemble sizes towards the coarsest resolution is used to guarantee the desired theoretical properties (Bierig and Chernov, 2014; Chernov et al., 2021). For a fixed work budget, optimal weights in the construction of the multi-level estimator could be derived (Schaden and Ullmann, 2020; Destouches et al., 2023) and strategies accounting for parallelisation should be considered (Mishra et al., 2012b).

In practical sequential data assimilation, a fixed multi-level ensemble size has to be chosen for all of the necessary estimation throughout the simulation. To ensure a proper assimilation quality, we choose the ensemble size tailored for the Kalman gain estimation in the MLEnKF. It is further expected that the correlation between the levels decreases over simulation time, so that we relate our choice to the last assimilation time in order to be conservative.

### 3.1.1 Practical vs. theoretic work

We draw the attention first to the scaling of computational costs over the hierarchy of grids. Even though the theoretical work required for solving the shallow-water equations is given by (2), numerical simulation software does not always give perfect scaling in practice, or at least not at for all problem sizes. Most often, small problem instances are not able to keep


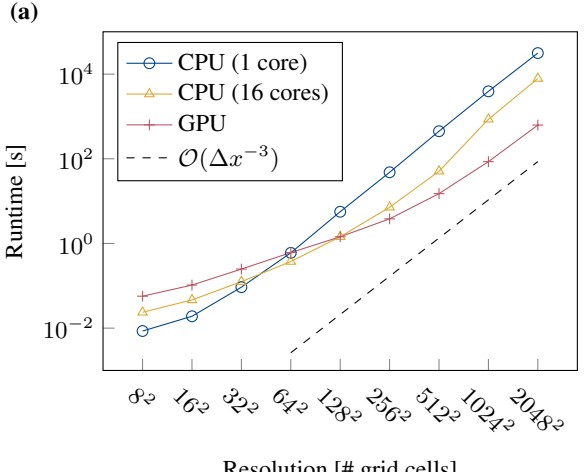
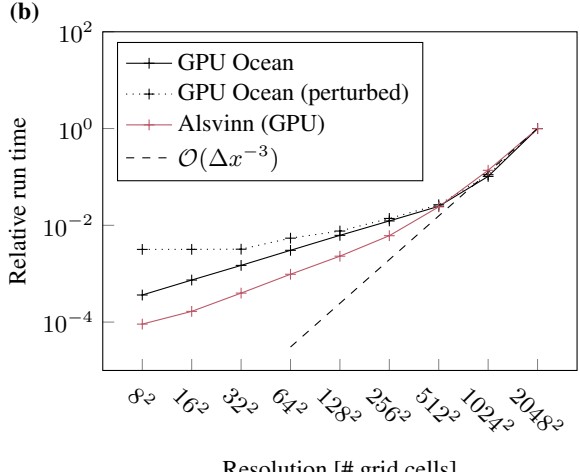

**Figure 2.** Practical computational work for different simulators with respect to number of grid cells. (a) Wall run time for Alsvinn on different computing infrastructures. (b) Relative run time compared to the largest problem instance for deterministic models using GPU Ocean and Alsvinn, and model with model errors using GPU Ocean. Both figures include theoretical scaling as a dashed line for reference.

the computer fully occupied, thereby leading to suboptimal computational performance. To investigate this issue, we consider two different simulation frameworks solving PDEs using finite-volume methods with explicit time stepping. The first one is the GPU-accelerated simulation framework GPU Ocean (Brodtkorb and Holm, 2021), which we will use to run simplified

ocean model in the later examples. The second one is Alsvinn (Lye, 2019), which is a more generic finite-volume framework specifically designed for MLMC applications, running both on CPU and GPU.

Figure 2 shows the computational performance of the two simulation frameworks with respect to the grid resolution. In Figure 2(a), we measure wall run times for Alsvinn using single-core CPU, multi-threaded CPU, and GPU. For large problems, we see that all these hardware configurations give perfect weak scaling (same slope as the dashed line) and that the GPU

code is roughly ten times faster than multi-core CPU and fifteen times faster than the single-core CPU code, respectively. However, the performance on the GPU drops more rapidly than on the CPU when the problem size decreases. This is because the very large number of computational cores on the GPU requires large problem instances to be fully utilised, and smaller problems lead to idle resources. The same pattern is also seen using a multi-core CPU. As a consequence, even though the GPU gives best performance on high resolutions, the single-core CPU performs best on extremely coarse grids. Alsvinn can

use the configuration that is best for a given grid resolution and moreover, pairs of simulations in the multi-level ensemble can be distributed between GPU for the finer partners and CPU for the coarser partners. As we observe that the run time of a high-resolution simulation on the GPU is about the same as the run time of a CPU simulation on the next coarser level, this heterogeneous parallelisation facilitates best possible use of the computational resources.





In Figure 2(b), we measure relative performance of GPU Ocean with and without additive model errors with respect to grid
resolution, keeping Alsvinn's performance on the GPU as a reference. GPU Ocean was developed for efficient high-resolution
simulation of simplified ocean models and we again see perfect scaling for problems with $> 512^2$ grid cells. For coarser
resolutions than $512^2$ we observe that both deterministic methods only achieve $\mathcal{O}(\Delta x^{-1})$ performance, which is expected when
a problem size can be executed fully concurrently on the GPU, such that only the time stepping becomes dominant. When we
perturb the model using additive model errors (see details in Sect. 4.1.2) every 60 s, however, we see that the performance drops
even more for coarse grid resolutions. Whereas the model perturbations are negligible at high grid resolution, it becomes more
and more dominating as coarser grids allow for larger finite-volume time steps. Eventually, the simulator alternates between
one numerical time step and one model error perturbation, and the computational cost does not reduce any further.

The potential speed-up through multi-level is the larger the steeper the slopes of the computational scaling, but based on the
observations in Figure 2, the assumption of perfect theoretical scaling cannot always be taken for granted. Although the GPU
simulations open up for only a mitigated speed-up compared to the single-core CPU simulations, it is beneficial for the over all
computational costs to use the simulation hardware that is fastest foe each given resolution. In the following, we will therefore
work with both assumptions, on the one hand with theoretical scaling as upper bound for the speed-up and on the other hand
with the practical costs to assess the actual speed-up for the GPU Ocean simulator.

### 3.1.2 Variance analysis

To take advantage of the theoretical results for MLMC discussed in Sect. 2.3.3, we must experimentally determine the values
of $v^l$ and $V^l$ for a function $g$. There exist various arguments for different functions and those arguments influence the choice
of the ensemble size implicitly. In the final forecasts, the mean is often the first quantity of interest, and hence one potential
choice for a function is

$$g_1(\boldsymbol{x}) = \boldsymbol{x}. \tag{20}$$

Alternatively, as the Kalman gain mostly depends on the estimation of the covariance matrix and the covariance has the form
$\mathbb{Cov}[\boldsymbol{x}] = \mathbb{E}[g(\boldsymbol{x})]$ with

$$g_2(\boldsymbol{x}) = \tilde{\boldsymbol{x}}^2, \tag{21}$$

we can use this definition of $g$ in the analysis.

To determine the values $v^l$ and $V^l$, we construct a trial experiment inspired by the corresponding MLMC trial experiments.
We start by defining $N$ ensemble members on every level, where one member per level shares the noise realisation with one
member on all the other levels. Then, we run a trial data assimilation experiment with a randomly sampled truth, where we
calculate the Kalman update on the finest level and project it to the coarser levels. Finally, we approximate the variances in
(16) for each physical variable individually.

In practice, it is prohibitive to run a trial experiment to determine the correlations between the levels for every new scenario
before running the multi-level data assimilation. We have repeated the same set-up with different stochastic truths and have





seen qualitatively very similar results, such that the hope is that one can extrapolate from one trial experiment to multiple scenarios. However, from a conservative mindset, the trial set-up has to be relevant enough for a new scenario, e.g., this is a reasonable assumption for tidal signals on different days, but not necessarily for other extreme weather conditions.

### 3.1.3  Optimal ensemble size

With the collected values for computational costs and variances per level, the multi-level ensemble size can be derived optimally in different ways. We follow the approach for optimal speed-up, such that for fixed statistical accuracy $\tau^2$, we set the ensemble size as

$$N^l = \left\lceil \sqrt{\frac{V^l}{C^l}} C_\tau \right\rceil \quad \text{with } C_\tau = \frac{1}{\tau^2} \sum_{l=0}^{L} \sqrt{V^l C^l}, \tag{22}$$

and choose $\tau^2$ as small as possible so that the resulting multi-level is still feasible within the computational budget (Müller, 355  2014). This is done per primary variable.

### 3.2  Localisation

Due to the finite sample approximation of the Kalman gain, spurious long-distance correlations are expected to degrade the quality of the updates. Localisation is a common strategy to improve the practical performance of EnKFs (Sakov and Bertino, 2010). Within MLEnKFs, localisation has also been analysed for level-local formulations (Hoel et al., 2020). However, we 360  will incorporate spatial localisation into the MLEnKF and the term "localisation" will in the rest of this article refer to spatial localisation. In the construction of the state covariance matrix, the entries corresponding to long-distance correlations are diminished and for the multi-level covariance estimator, we can pick this tapering for each level individually (Destouches et al., 2023).

Let $\mathbf{w} \in \mathbb{R}^{n_x \times n_y}$ be a tapering matrix that contains identical rows $w$. In the updates in (9) or (19), we use the Schur product 365  of the kernel and the Kalman gain $\mathbf{w} \circ \mathbf{K}$. Generally, the need for localisation primarily arises due to a small number of ensemble members. As the involved number of ensemble members varies per level in the multi-level case, the MLEnKF offers the possibility to include localisation on each level independently. We incorporate this directly into the construction of the Kalman gain by

$$\boldsymbol{\Sigma}_{XY}^{\text{ML,loc}} = \frac{\mathbf{w}^0}{N^0} \circ \sum_{e=0}^{N^0} \tilde{\boldsymbol{x}}_e^0 (\tilde{\boldsymbol{y}}_e^0)^\top + \sum_{l=1}^{L} \frac{\mathbf{w}^l}{N^l} \circ \sum_{e=1}^{N^l} \left( \tilde{\boldsymbol{x}}_e^{l+} (\tilde{\boldsymbol{y}}_e^{l+})^\top - \tilde{\boldsymbol{x}}_e^{l-} (\tilde{\boldsymbol{y}}_e^{l-})^\top \right), \tag{23}$$

where $\mathbf{w}^l$ entries are either $w$ or 1. In this case, $\boldsymbol{\Sigma}_{YY}^{\text{ML}}$ cannot be extracted from $\boldsymbol{\Sigma}_{XY}^{\text{ML,loc}}$ as in (8b), but has to be calculated on its own by (14b). The level-wise application for localisation leads to a biased estimator in theory, but localisation has shown to be a powerful heuristic in practice (Carrassi et al., 2018). For the case of spatially sparse observations, $n_y \ll n_x$, one can also conduct sequential processing of observations and choose $w$ including relaxation (Beiser et al., 2023).





### 3.3 Negative eigenvalues

While the empirical measure associated with a classical Monte-Carlo estimator is positive in the sense that all Dirac contributions have a positive sign, this is not the case for multi-level estimators where the differences between the levels can introduce negative values. For example, the lack of the positiveness property yields that approximated variances can become negative, which is statistically invalid. Moreover, the multi-level covariance estimator is no longer guaranteed to be positive semi-definite, which can be seen from the estimated covariance matrix having negative eigenvalues.

Methods to construct positive semi-definite matrices are expensive (Maurais et al., 2023) and the calculation of eigenvalues for large matrices can be prohibitively costly. We circumvent the assemblance of the full covariance matrix $\Sigma_{XX}$ and rather work with the low-dimensional matrix $\Sigma_{YY}$ in the suggested MLEnKF approach. With very sparse observation data, its eigenvalues are fast to compute.

While developing the code for the numerical experiments in Sect. 4, we noticed rare occasions of slightly negative eigenvalues and we identify these to be associated with situations where the observation locations lie on a wave front. If no mitigation strategy is implemented and $\mathbf{K}$ is used directly in the update, the ensemble shows non-physical behaviour over time and eventually becomes useless. Visually, strong waves in opposite directions take over to dominate the dynamics. For plain MLMC this is less of a problem, but it becomes pivotal for data assimilation within the MLEnKF and requires special attention. Since this seems to happen only rarely in our experiments, we overcome this issue simply by skipping the data assimilation to avoid 390 non-physical properties in the multi-level estimation, when the smallest eigenvalue $\Sigma_{YY}$ becomes negative. By doing this, we have noticed that the ensemble results stayed well calibrated and physically reasonable. We monitored slightly negative eigenvalues in less that 1:10.000 times of the assimilated observations, such that missed assimilation becomes negligible.

Alternatively, one could replace the Kalman gain by a single-level Monte Carlo estimator from one of the levels in case of negative eigenvalues, but we have seen that it is most important to keep updates statistically and physically sound. In mitigation 395 strategies, it is not enough to manipulate only $\Sigma_{YY}$, but also $\Sigma_{XY}$ should be corrected correspondingly as we have learnt from independent experiments.

## 4 Application to simplified ocean model

We now study the performance of MLEnKF for assimilating sparse observations into a simplified ocean model. The experiment setup is chosen to model rotational phenomena that resembles ocean currents, and the intention is to first use the observations 400 from our synthetic truth to do state estimation (in this section), and then use the posterior multi-level ensemble to forecast drift trajectories (Sect. 5).

### 4.1 Problem description

As our numerical test case, we consider two geostrophic jets in a periodic domain modelled by the shallow water equations in a rotating frame of reference. Although the jets initially represent steady state, small perturbations lead to chaotic instabilities,



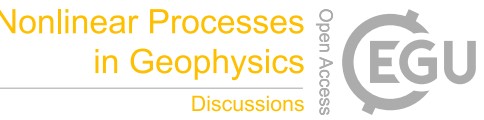

meaning that pure Monte Carlo runs without any data assimilation result in a symmetric multivariate distribution with large spread around the mean. We consider an identical-twin experiment, where we run an independent simulation which represents the truth, and from which we sample observations. We then use the same model to run the multi-level ensemble, and use MLEnKF to assimilate the observations. In general, and unless otherwise stated, our experiment consists of three phases:

   – three days of model spin-up,

– seven days of data assimilation, and

   – three days of forecast.

This specific experimental design have previously also been used in both Holm et al. (2020b) and Beiser et al. (2023), although with different configurations for the observations.

### 4.1.1   Shallow-water model and initial conditions

The shallow-water model is a non-linear hyperbolic system of equations for conservation of mass and momentum in two dimensions given by

$$
\begin{bmatrix} \eta \\ hu \\ hv \end{bmatrix}_t + \begin{bmatrix} hu \\ hu^2 + \frac{1}{2}gh^2 \\ huv \end{bmatrix}_x + \begin{bmatrix} hv \\ huv \\ hv^2 + \frac{1}{2}gh^2 \end{bmatrix}_y = \begin{bmatrix} 0 \\ fhv \\ -fhu \end{bmatrix}. \tag{24}
$$

Here, $\eta$ represents the sea surface elevation as a deviation of some equilibrium mean, and $hu$ and $hv$ are the moments in $x$- and $y$-directions. The equilibrium depth is given by $H$, so that the total water depth becomes $h = H + \eta$. Furthermore, we have the

gravitational constant $g = 9.81\mathrm{m/s}^2$ and the Coriolis parameter $f = 0.0012/\mathrm{s}$. To represent the model $\mathcal{M}$, we simulate (24) using the GPU-accelerated framework GPU Ocean (Brodtkorb and Holm, 2021), which uses a high-resolution central-upwind finite-volume scheme based on Chertock et al. (2017). For the numerical experiments, we consider a rectangular domain of size $666\mathrm{km} \times 1332\mathrm{km}$ with constant depth $H = 230\mathrm{m}$ and periodic boundary conditions. At the finest resolution we span a $512 \times 1024$ grid such that the state vector $\boldsymbol{x}^L = (\eta_{i,j}, hu_{i,j}, hv_{i,j})_{i=1,j=1}^{512,1024}$ collects all the conserved variables within the grid.

The initial conditions consists of two opposed jets in the northern and southern part of the domain, and is motivated by a test case suggested by Galewsky et al. (2004) for validating shallow-water models on a rotating sphere. These jets are initially in a steady-state with respect to geostrophic balance, meaning that they satisfy the relation

$$
hu = -\frac{gh}{f}\frac{\partial \eta}{\partial x} \quad \text{and} \quad hv = \frac{gh}{f}\frac{\partial \eta}{\partial y}. \tag{25}
$$

The system is however unstable, meaning that small perturbations lead to developments of complex currents and eddies over

time. By introducing perturbations through random sampling of the model error term, the currents of different model realisations will evolve in different ways and the model behaves chaotically.





### 4.1.2 Model perturbations

The random model errors $\delta\boldsymbol{x}$ collects perturbations $\delta\eta, \delta hu, \delta hv$ for each of the physical variables. We construct the noise by generating samples from a Karhunen-Loeve-type random field for $\delta\eta$, and assigning $\delta hu$ and $\delta hv$ according to the geostrophic balance in (25). With a scaling parameter $\Theta$ and a decay parameter $\theta$, the perturbation in $\eta$ is created by

$$\delta\eta = \Theta \sum_{i=i_0}^{i_0+I} \sum_{j=j_0}^{j_0+J} \alpha_{i,j}^{\sin} \, i^{-\theta} j^{-\theta} \sin(2i\pi\zeta) \sin(2j\pi\xi) + \alpha_{i,j}^{\cos} \, i^{-\theta} j^{-\theta} \cos(2i\pi\zeta) \cos(2j\pi\xi). \tag{26}$$

We here parameterise the domain as $\zeta, \xi \in [\alpha^x, 1+\alpha^x] \times [\alpha^y, 1+\alpha^y]$ and introduce the shifting $\alpha^x, \alpha^y \sim \mathcal{U}[0,1]$ to break up spatial patterns. Note that $\delta\eta$ is continuous and it is evaluated in the cell centres to get discrete realisations. For the rectangular domain, we set $i_0 = 1$ and $j_0 = 2$ to obtain basis functions with the same physical scales in $x$ and $y$ directions. Moreover, we use $I = J = 7$ basis functions per axis, $\theta = 0.9$ for the decay, and $\Theta = 0.001$ for the scaling.

For multi-level methods, it is essential that the partners in a pair share the same realisation. When the stochasticity enters the system only via initial conditions or external forcing, the coupling is rather easy to realise in the implementation. However, in this MLDA case, we have frequent perturbations working directly on the state variables, which means that the two partner simulations should share the same realisation of the model error on their different levels. It is thus necessary to evolve coupled simulators to the same time and transfer information about the sampling. Since the performance of the simulation of a coupled pair should not be affected much by the model error, we take care that only some data for the sampling is transferred between the partners.

The necessary information to sample a single realisation of $\delta\boldsymbol{x}$ is contained in the $2IJ$ random numbers $\alpha_{i,j}^{\sin}, \alpha_{i,j}^{\cos} \in \mathcal{U}[-1,1]$ and $\alpha^x$ and $\alpha^y$. Therefore, the same realisation of the model error can be easily applied to coupled simulations on different levels by communicating only these numbers and evaluating (26) in the cell centres of the corresponding grid. For all numerical experiments, we employ a model time step $\Delta t = 60$s meaning that we perturb the simulated states after every minute.

### 4.1.3 Sparse in situ observations

After $3$d of spin-up, we retrieve observations from the synthetic truth simulated on the finest level for the next $7$d. In particular, the truth is perturbed with the same model error distribution as the ensemble and is displayed in Figure 3. We see the turbulent evolution starting from the steady state at $t = 0$d. We consider 50 regularly distributed locations, that are about $100$km apart from each other, for momentum observations and we collect data every $15$min. Note that $\eta$ is unobserved. For the observation noise, we choose a standard deviation of $0.1$m/s in the current velocities, which corresponds to $\mathbf{R} = 500\mathbf{I}_2$ as the observation covariance matrix for the momentum at a single location. Furthermore, we parameterise the localisation with a radius of $50$km and a relaxation factor $0.25$, where details on the localisation radius and the relaxation can be found in Beiser et al. (2023).

### 4.2 Multi-level configuration

To first get an impression of how the relevant dynamics are maintained by different grid resolutions in our case, we run a single realisation of the model on different grid resolutions, where all of them use the same realisation of the model error. Figure 4


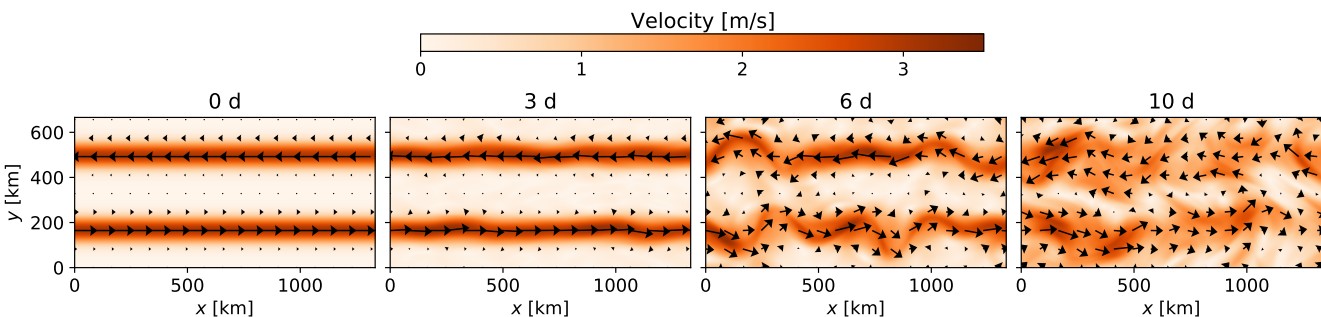

**Figure 3.** The synthetic truth at different times during the experiment: Starting with a westward jet in the north and an eastwards in the south, turbulent dynamics builds up over the following days and becomes clearly visible from day 6.

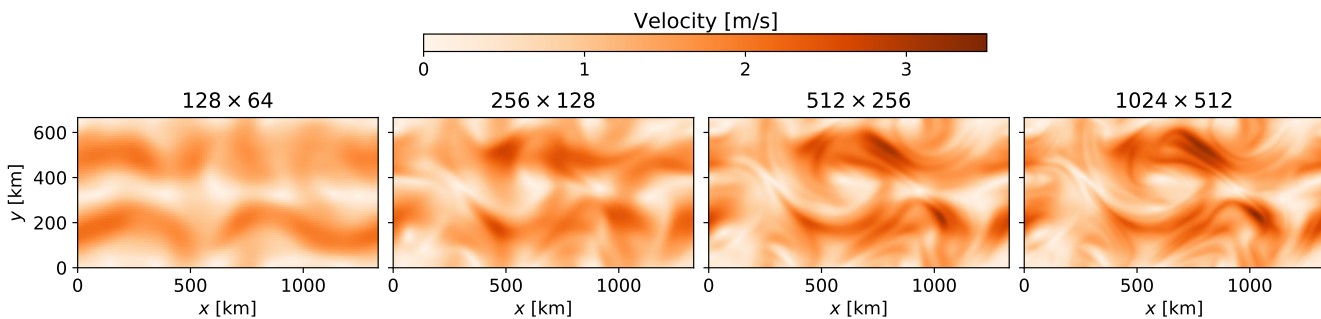

**Figure 4.** Same realisation of the shallow-water model for different resolutions after simulation until $t = 10$d.

shows the resulting currents at $t = 10$d. Here, we see that on the two finest levels, $1024 \times 512$ and $512 \times 256$, the currents are almost the same with only minor details on the sharpness of some features. With $256 \times 128$ grid cells, the main structures are still intact, although most of the details are smoothed out. On the coarsest resolution $128 \times 64$, however, we no longer recognise the shape of the currents from the finest resolution, and the solution contains few details in general.

To find the optimal configuration of the multi-level ensemble sizes as discussed in Sect. 3.1, we need to experimentally assess the computational cost of running the model at each level $c^l$, the variance at each level $v^l$, and the difference between the variance for partner levels $V^l$. For the GPU Ocean code, we saw in Sect. 3 that we only maintain theoretic computational performance for problem instances with more than $512^2$ grid cells. We therefore measure the practical cost of running the model for the grid sizes that we will use in our multi-level experiments to determine the computational work $C_l$. These values behave qualitatively very similar to the curve in Figure 2 where the considered resolutions are in the range where we are close to the theoretical scaling between the two finest levels, but the curve flattens out for the coarser levels. Because the relative high computational cost for coarse simulations is somewhat special for GPU-accelerated code, and we also want to discuss the practicability of multi-level data assimilation in general, we will consider both the theoretical cost and practical cost

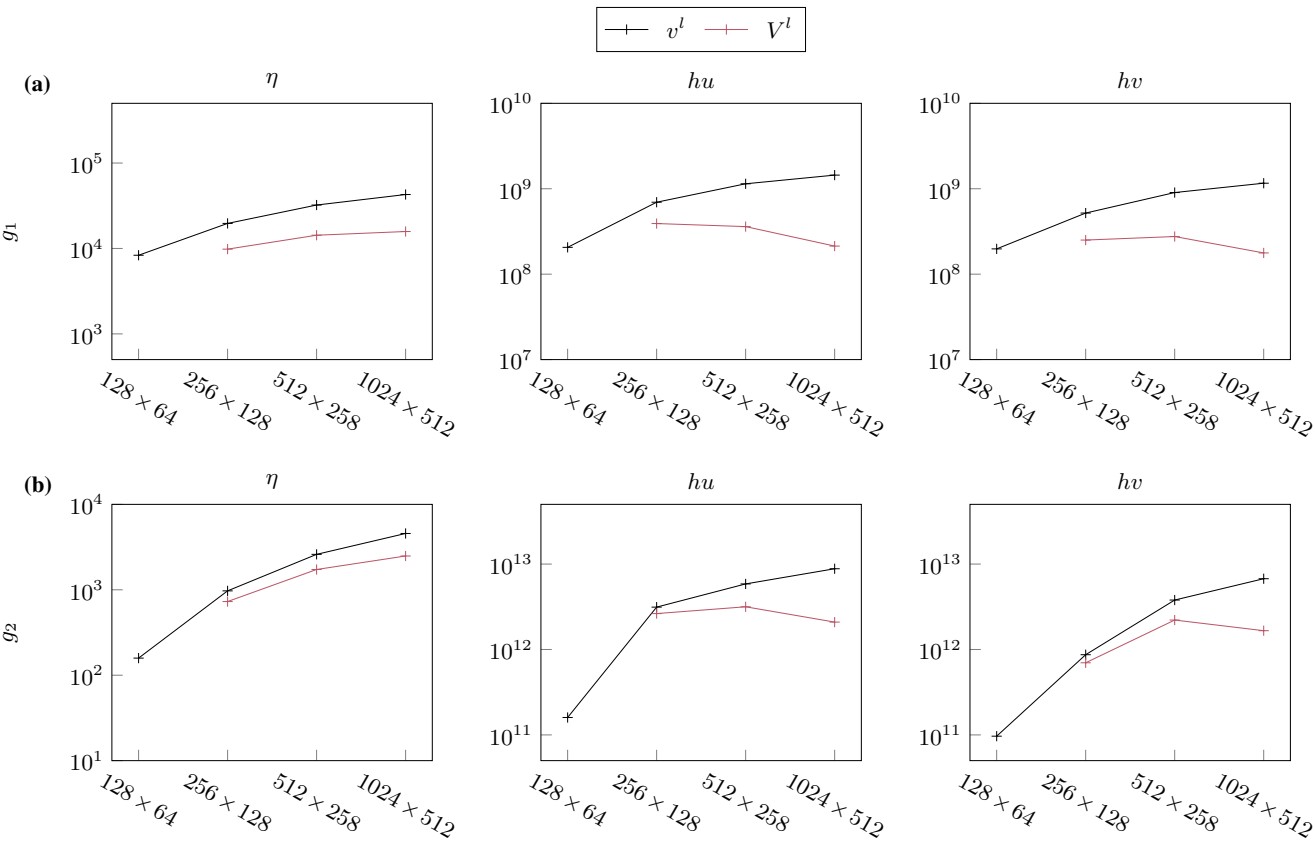

**Figure 5.** Variance analysis per variable: Showing the variance per level (blue lines) and between the partner (red lines) after 10 days of trial experiment for (a) $g_1$ and (b) $g_2$, where $g_1$ represents the mean as quantity of interest and $g_2$ the variance, respectively.

when configuring the multi-level ensemble size. The assumption of theoretic computational cost then represents the speed-up potential through using multi-level statistics if obtainable through other software frameworks.

Figure 5 shows the results of a trial experiment as in Sect. 3.1.2 for quantities of interest based on the mean $g_1$, as given by (20), and the covariance $g_2$, as in (21). Hereby, the variances are recorded per variable, such that the columns in the display correspond to $\eta$, $hu$ and $hv$. In all panels the curve for $v^l$ is increasing, indicating that many details are lost at the coarser levels. Also, all curves for the variances between the levels $V^l$ lie below $v^l$, but they do not have a strongly decreasing slope. For the function $g_2$ in panel (b) both curves are even closer together than for $g_1$ in panel (a). We note that the relative changes between the levels is most relevant for the ensemble size and not the actual magnitudes. In the data assimilation experiment, only data for the momentum was assimilated and the plots for $hu$ and $hv$ are qualitatively similar per function, but for $\eta$ it looks different.

While this is not an ideal scenario for maximal speed-up through multi-level estimators, these kinds of variances are more likely to be faced in many practical scenarios. In this sense, the case is on the edge for the multi-level approach to be beneficial.





**Table 1.** Ensembles with the same theoretical error as a single level ensemble with $N_e = 50$ in relation to the listed functional.

| Case Name | Work for $C^l$ | Function for $V^l$ | SWE variable | Ensemble Size | | | | Speed Up Factor |
|-----------|----------------|--------------------|--------------|---------------|---|---|---|-----------------|
| | | | | $1024 \times 512$ | $512 \times 256$ | $256 \times 128$ | $128 \times 64$ | |
| (A) | theoretical | $g_1$ | $hu$ | 13 | 46 | 134 | 275 | 2.3 |
| (B) | practical | $g_1$ | $hu$ | 14 | 86 | - | - | 1.7 |
| (C) | theoretical | $g_2$ | $hu$ | 19 | 66 | 185 | - | 1.7 |

For a speed-up through multi-level estimation, it would be beneficial if the fine levels have very strong correlation. If the simulation results on the finer levels become too similar, however, there would be no incentive to use very fine resolution, as a coarser, and thereby less costly, level can deliver the same information. In such a case, the limitation often lies in the model and no longer in the resolution. Nevertheless, if fine grids are important for the resolution of local features in parts of the domain, multi-level methods are still meaningful despite that those high resolutions may be strongly coupled globally.

The optimal ensemble sizes depend on the number of levels, the computational cost per level $C^l$ and the function $g$. The latter goes into expression (22) via the variances $V^l$. For the computational costs, we consider configurations with both theoretical scaling and the practical performance. Because $hu$ is dominating in the assimilation for most of the experiment time, we pick this primary variable for further analysis. We then choose multi-level ensemble sizes that match the theoretical error of a single-level ensemble with 50 members for the same function. Table 1 lists three different multi-level configurations along with their corresponding speed-up. To have a representative set of sufficiently different experiments, we chose a different number of levels for each of the cases. Either way, we note that in all cases the ensemble sizes increases towards the coarser levels. The computational work of the single-level ensemble is $\mathcal{C}^{SL} = 50\, c^L$ and the computational work of the multi-level ensemble $\mathcal{C}^{ML}$ is calculated by summing over the levels analogously, meaning that the speed-up is a factor of

$$\frac{\mathcal{C}^{SL}}{\mathcal{C}^{ML}}, \tag{27}$$

where the computational costs for the data assimilation are neglected in this consideration. It is crucial to note that multi-level methods do not always yield a speed-up and that this depends on the characteristics of the problem represented by $C^l$ and $V^l$. For the combination of practical work and function $g_2$, we will not obtain a speed-up and have therefore not included that combination.

### 4.3 Numerical results

Such a complex example as in this article is to the best of the authors knowledge not yet covered by any theoretical results. Moreover, the multi-level ensemble is derived for the estimator of certain quantity of interest, but there is no mean-field EnKF distribution accessible for practical comparison. For understanding the properties of practical data assimilation methods, one is often more interested in the prediction error versus the data or the simulated truth.



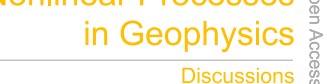
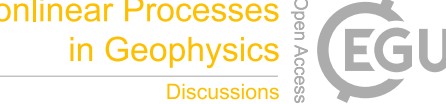

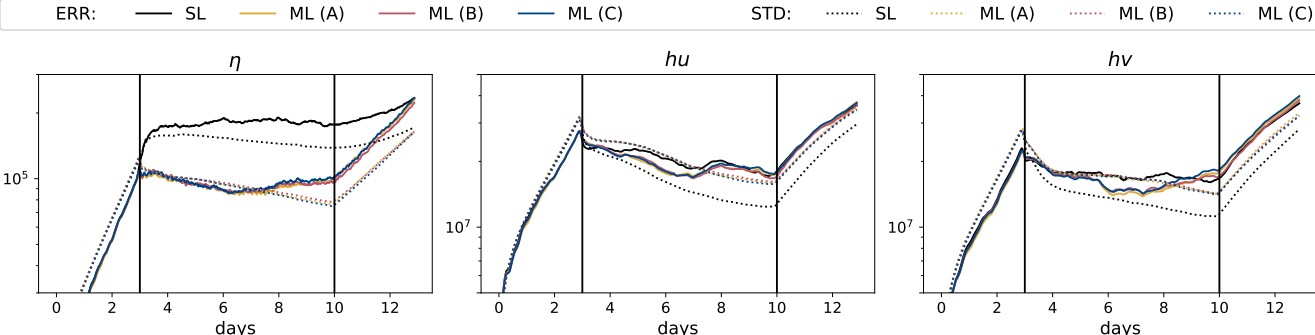

**Figure 6.** ERR and STD during the data assimilation experiment for ensembles of equal theoretical error. The curves are smoothed with a moving average over a 3h interval to reduce the noise in the plot. Vertical lines mark transitions between spin-up, data assimilation, and forecast.

### 4.3.1 Calibration and spread

We assess the error in the mean (ERR) of the ensemble per variable

$$\mathrm{ERR} = \|\boldsymbol{\mu}[\boldsymbol{x}] - \boldsymbol{x}_{\mathrm{true}}\|_2, \tag{28}$$

where the mean estimate comes from a single-level ensemble as in (5) or from a multi-level ensemble as in (14). Note that the
criterion in (28) was not directly used in the design of the multi-level estimators, but it gives insight about the state estimation resulting from the data assimilation. We oppose the ensemble calibration with the ensemble spread measured by the empirical standard deviation (STD).

In Figure 6, we show the ERR (solid) and STD (dotted) over time for four different experiments for each of the physical variables. Keep in mind that the phase before 3d is spin-up and that after 10d is plain forecast. The black lines represent a
single-level experiment with $N_e = 50$, whereas the other three represent the three multi-level configurations from Table 1. For the single-level experiment, we see that the STD lies under the ERR, but it follows the same trends. For the multi-level experiments, they all show the same quality in terms of ERR for $hu$ and $hv$ (middle and right display), confirming that the assumption of same statistical errors in the ensembles was legit. For the unobserved variable $\eta$ (left display), the error for the single-level ensemble on the finest level is quite noisy, but the multi-level ensemble smooths this out. This is a feature of
multi-level ensemble methods, as it benefits from a coarse estimation with less spurious correlations, but we do not want to overrate this as it does not have a big impact on the currents. In our experiments, the assumption of the different computational costs per level does not affect the results, as expected. More surprisingly, the use of the interchanging of $g_1$ and $g_2$ does not change the curves much either. All results for ERR of the moments are basically equivalent, but the STD is a bit higher for the multi-level cases. Due to high costs of a single experiment in this practical setup, it is prohibitive to evaluate the statistical
error by repetitions of the full experiment.


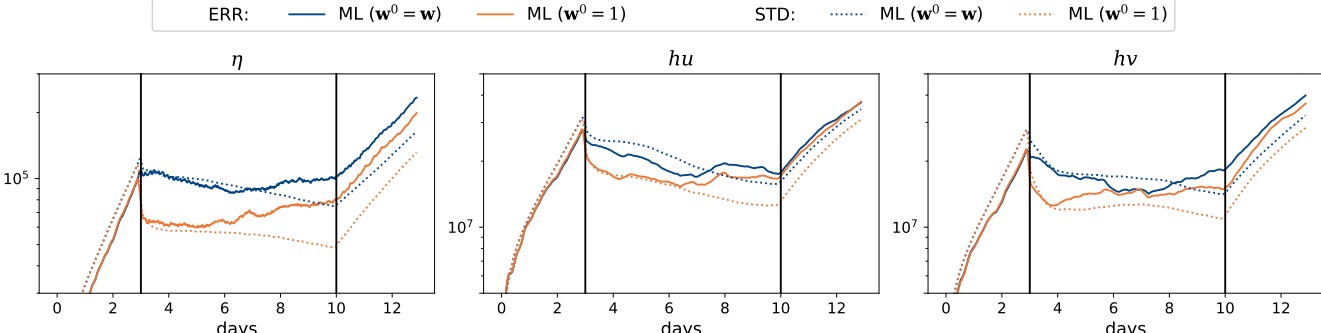

**Figure 7.** ERR and STD during the data assimilation experiment for the multi-level ensemble (A) with (blue) and without (red) localisation on the coarsest level.

While it is not practical to abstain from localisation with only 50 ensemble members in the single-level case, it is reasonable to do so for the coarsest level of case (A) with 275 members. In Figure 7, we show the comparison of localisation on all levels (blue) and localisation on all but the coarsest level (yellow). There is no equivalent single-level experiment for this approach. Comparing the ERR for the two experiments, we clearly see that the ensemble without localisation on the coarsest level is

pushed much stronger towards the truth in the beginning of the DA period. This is because the updates are not tapered but rather extend over the entire domain. Together with this, the standard deviation is also reduced by a similar magnitude in the early phase. However, once the dynamics becomes more turbulent around day 6 to 7, the curve for ERR without localisation on the coarsest level approaches the other experiment. In this unstable regime, small spurious correlations seem to be enough to cause deviations from the truth. The ensemble without localisation at the coarsest level keeps the lower STD during the

remaining course of the experiment.

### 4.3.2    Multi-level eligibility

As discussed before, MLDA differs from MLMC by the fact that the ensemble is repeatedly used for estimation over time. In Figure 8, we plot the relative variance of the difference in the function $g$ at level $l$ compared with the variance of $g$ at that level, e.g. $V^l/v^l$ defined in (16). Going from dark blue (fine level) to light blue (coarse level), the variance in the difference gets

relatively larger compared with the variance in the function. The dotted lines represent the relative variance in a pure Monte Carlo experiment without any data assimilation, whereas the solid lines show the situation with data assimilation using the MLEnKF. In all three displays ($\eta$, $hu$, $hv$), the solid lines tend to go above the dotted line when data assimilation starts after $3\mathrm{d}$ of spin-up. Data assimilation reduces the uncertainty in the ensemble, and this means the values for $v^l$ (denominator here) will go down, especially in the beginning of the data assimilation process. The exception is the coarsest level, where the solid and

dotted lines are much more similar. This indicates the sparse observations has most effect locally, and the relative differences are hence largest at the finer levels.


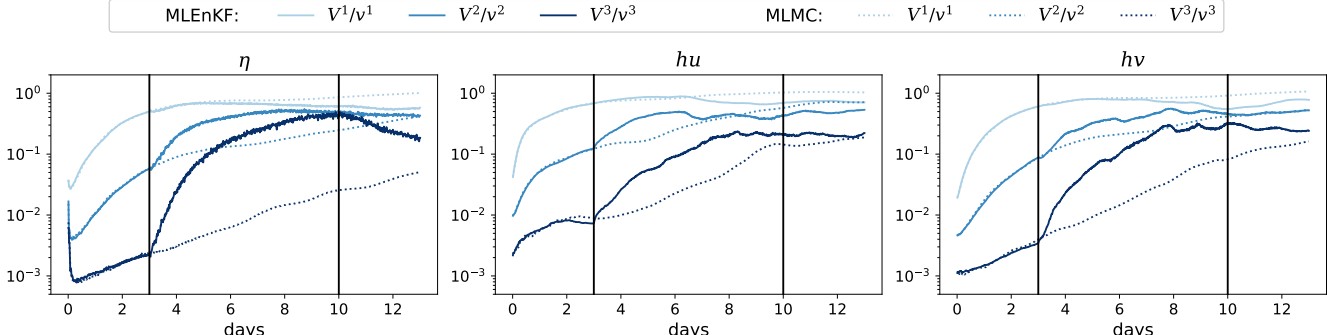

**Figure 8.** Relative variance of the level differences to the variance on those levels over the time of the data assimilation experiment (A). MLMC experiment with same ensemble size for reference.

### 4.3.3 Rank histograms

To assess the spread in the analysis ensemble produced by the MLEnKF, we repeat a shorter version of the data assimilation experiment multiple times. We choose a multi-level ensemble of sizes 12, 37 and 88 on the three highest levels, corresponding to a single-level ensemble with 40 members for $g_1$ and practical work. For each of these replicate experiments, we initialise a new truth and a new ensemble using the same parameters as before. We then spin the ensemble up for three days, followed by one day with data assimilation and three hours of forecast. At the end of the forecast, we record the truth and the ensemble values at 24 different locations in the domain. These locations are chosen by a regular equidistant $4\times6$ pattern, and are assumed to be uncorrelated within the time frame of the assimilation and forecast.

Rank histograms are a common technique to visually evaluate sample and ensemble spread, where uniform histograms are usually considered desirable (Hamill, 2001). For multi-level ensembles, the formulation by a probability integral transform value is useful (Gneiting et al., 2007), where one rank corresponds to the truth's function value of the empirical cumulative distribution function represented by the ensemble. The rank can therefore be estimated as an expectation value. For the true observation value $y$, the multi-level rank at an observation location can be defined by evaluating the multi-level estimator in (13) with the indicator function $g = 1_{[-\infty, y]}$. The ranks are then given as continuous values, meaning that we can choose the histogram bins independent of the ensemble sizes. Similarly to the practical challenges discussed in Sect. 3.3, a straightforward evaluation of the indicator function does however lead to some imperfections caused by the multi-level estimator. First, the ranks are not guaranteed to be bounded between 0 and 1, and second, the multi-level ensemble is not optimised for the approximation of this particular discontinuous function.

Figure 9 visualises the histograms for 138 repeated rank experiments using bins of width 0.05. Altogether, this means $138 \cdot 24 \approx 3000$ testing observations. Most prominently, we see that the multi-level rank histograms differ from classical single-level histograms by taking values smaller than 0 and bigger than 1. However, in this case, those values are the minority and the rank histograms can be interpreted similar to classical ones. The histogram for $hv$ is close to a uniform distribution indicating

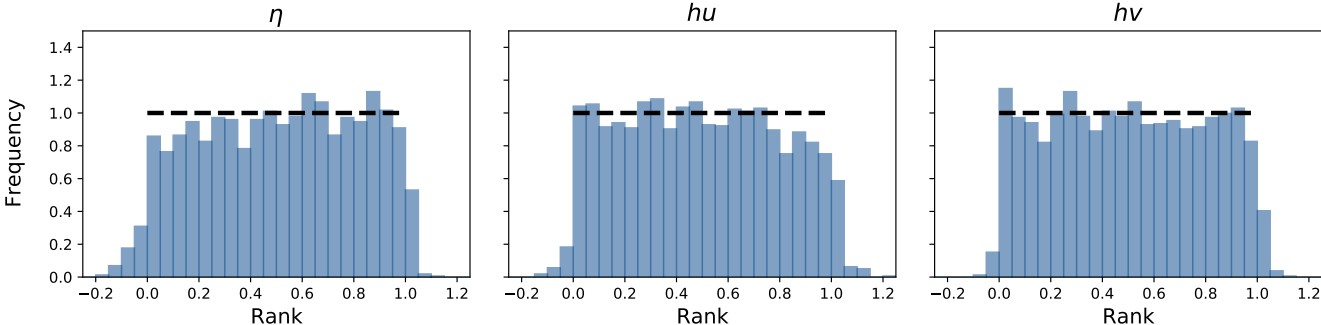

**Figure 9.** Multi-level rank histograms for each primary variable generated by more than 3000 observations. The dotted line represents a uniform distribution for reference.

that the statistics of the multi-level posterior ensemble is correct in the sense that it has the same distribution as the truth in

this experiment. Also for $hu$, we recognise neither a clear bell nor U-shape. There are however fewer ranks associated with the larger values, but at the same time we also have the largest collection of values that are larger than one. While the histogram for $\eta$ has more outliers in the lower end, the general rank histogram is still close to uniform, such that we see no indication of systematic errors in the data assimilation with the MLEnKF.

## 5 Multi-level drift trajectory forecasting

In the previous section, we explored the multi-level state estimation problem through data assimilation, and we assessed the quality of the approximate posterior distribution of the ocean states directly. Now, we discuss how we can use the multi-level ensemble to forecast derived quantities of interest, specifically through probabilistic forecasting of drift trajectories.

### 5.1 Multi-level drift trajectory model

Drift trajectories are typically computed either online as part of the dynamical ocean model, or offline through dedicated drift

trajectory simulation software that takes the output files from the dynamical ocean simulator as input (Dagestad et al., 2018; Delandmeter and van Sebille, 2019). Here, we consider the online approach using a passive drift model. We let $\Psi(t) \in \mathbb{R}^2$ contain the drifter's $x$ and $y$ coordinates at time $t$, and update its position by

$$\Psi(t + \Delta t) = \Psi(t) + \Delta t \begin{bmatrix} u(t, \Psi(t)) \\ v(t, \Psi(t)) \end{bmatrix}. \tag{29}$$

Here, the time step is the same as the time step for the simplified ocean model, and $u$ and $v$ are bi-linear reconstructions of the

simulated ocean currents in space to best represent the exact location of the drifter at time $t$.

Using a single-level ensemble of dynamical models, the simplest way to produce a probabilistic trajectory forecast is to simulate one drifter per ocean ensemble member and solve (29) for each of them. In a multi-level ensemble, however, the





individual ocean states are no longer realisations from the statistical distribution they collectively describe, and the "one drifter per ocean model" approach fails. Instead, we must construct an ensemble of $N_d$ drifters that behave according to the statistical

estimates of the currents obtained from the multi-level ensemble. One such option is to let each drifter realisation be advected according to a random walk by sampling

$$u_d(t, \Psi_d(t)) \sim \mathcal{N}(\mu_u(t, \Psi_d), \sigma_u(t, \Psi_d(t))), \tag{30}$$

where the normal distribution mean $\mu_u$ and standard deviation $\sigma_u$ are based on that of the multi-level ensemble using (14) for the velocity $u$ at time $t$ and position $\Psi_d(t)$ for drifters $d = 1, ..., N_d$. A corresponding expression is used for $v$.

If the sampling in (30) is uncorrelated in time, however, all drifters will be estimates of the mean as $t$ increases. This can be seen through a comparison to the single-level "one drifter per ocean model" approach. In the single-level case, a drifter that is associated with an ocean state that represents an extreme from the ocean state probability distribution will consistently generate a corresponding trajectory that becomes an extreme in the trajectory distribution as well. By sampling the currents according to (30), the probability that one drifter consistently follows a extreme current is highly unlikely.

For this reason, we instead simulate drifters through what can be described as a *biased walk*. Thus, we let $\beta_{d,u}, \beta_{d,v} \sim \mathcal{N}(0, 1)$ be a random normal distributed bias for the velocities $u$ and $v$, respectively. We sample these only once for each drifter $d = 1, ..., N_d$ and keep these fixed over time. We then evolve each drifter's position by

$$\Psi_d(t + \Delta t) = \Psi_d(t) + \Delta t \left[ \begin{array}{c} \mu_u(t, \Psi_d) + \sqrt{\beta_{d,u}} \sigma_u(t, \Psi_d(t)) \\ \mu_v(t, \Psi_d) + \sqrt{\beta_{d,v}} \sigma_v(t, \Psi_d(t)) \end{array} \right], \tag{31}$$

so that each drifter follows a consistent path within the probability distribution for the currents. Note that (31) also can be used

for a single-level ensemble by using the corresponding single-level estimators for the velocities.

## 5.2  Numerical results

In the following, we generate three days trajectory forecasts for passive drifters starting at 100 randomly chosen initial locations in the domain at day 10 of our experiment as described in Sect. 4.

We consider three different cases: Classical single-level trajectories generated by attaching drifters to each ocean model

ensemble member; using the statistical approach of the biased walk in (31) for exactly the same single-level run; and using the biased walk for the multi-level ensemble experiment from case (A) in Sect. 4. For the biased walks, we choose $N_d = 50$ drifters and $\Delta t = 60$s.

Figure 10 shows the trajectory forecast with the three different methods for three different initial positions. For drifter (a) in the top row, all drifter ensembles follow the curve of the true trajectory and cover the true drifter's final position with slightly

different spread. Drifter (b) is harder to predict as all three methods suggest that it can either drift down to the right or up to the left. We see, however, that the multi-level ensemble suggests higher probability to the left which is correct, whereas both methods based on the single-level ensemble give higher probability to the right. On the other side, the true trajectory for drifter (c) in the lowermost row is only an outlier in the the multi-level ensemble. Here, the single-level ensemble gives a good

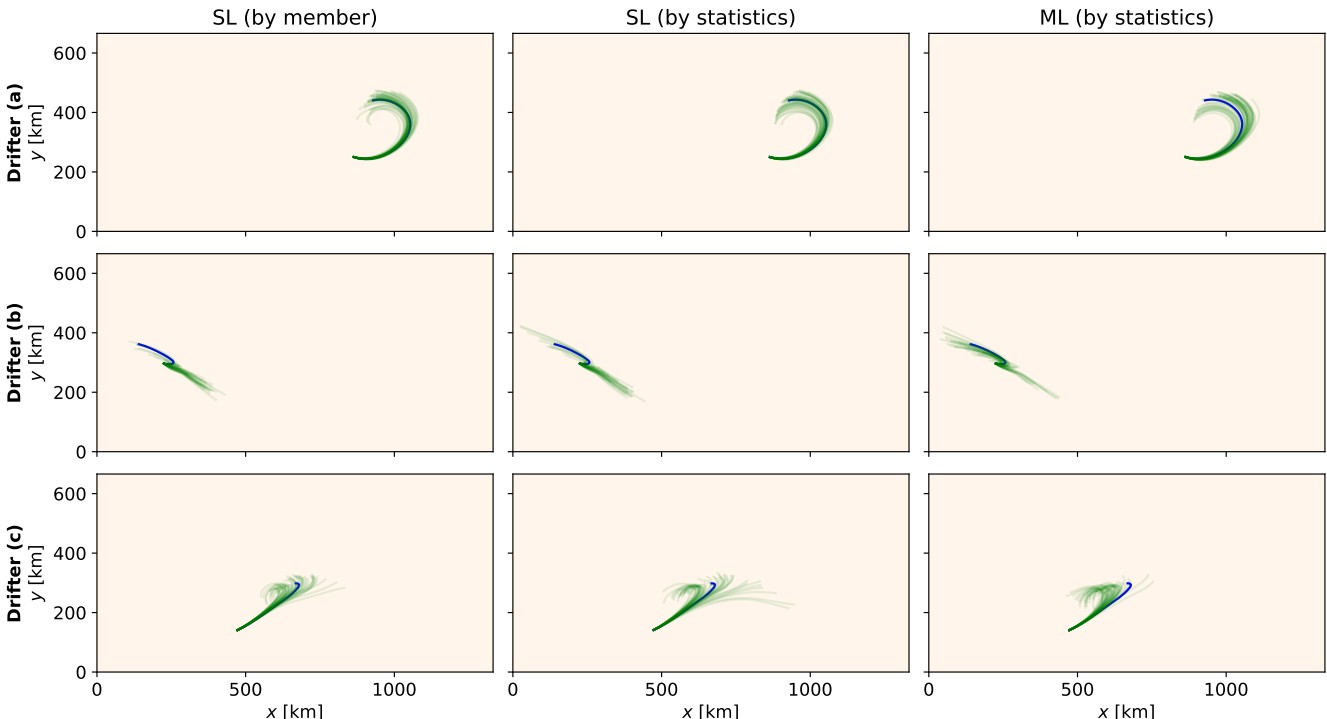

**Figure 10.** Drift trajectories for single-level and multi-level ensembles with different trajectory modelling approaches. We use 50 ensemble members in the forecasts by statistics, and for the multi-level plot, we use the ensemble of case (A). The blue trajectory comes from the true realisation. We show the trajectories for three different starting positions.

prediction of the true drifter when we make the forecast directly with one drifter per ocean ensemble member. Whereas for the biased walk, the true trajectory is still well represented by the ensemble, the ensemble has a very large spread.

This shows that it is hard to assess the quality of each method based on the trajectory forecast of just the a few initial positions. We therefore look at the ensemble calibration and spread for all 100 drifters similar to what was done for the state estimation. During the drifter forecast period in the experiment, the drifter error in the mean is

$$\text{ERR}(t) = \|\mu[\Psi(t)] - \Psi_{\text{true}}(t)\|_2. \tag{32}$$

In Figure 11, the drifter error ERR and the standard deviation of the drifter ensemble (STD) is averaged over all 100 starting positions and plotted over time for the different methods. We notice that the calibration curves for all methods are very close, but the spread using multi-level methods is a little bit bigger. This is in line with the results of the state estimation in Figure 6. The same single-level ensemble is used twice here for both trajectory modelling approaches, such that these differences originate not in the ocean states but purely from the drifter modelling. Here, we see that the spread of the biased walk is larger than of the traditional method per ensemble member. Hence, we can pinpoint the increased spread of the multi-level drifter



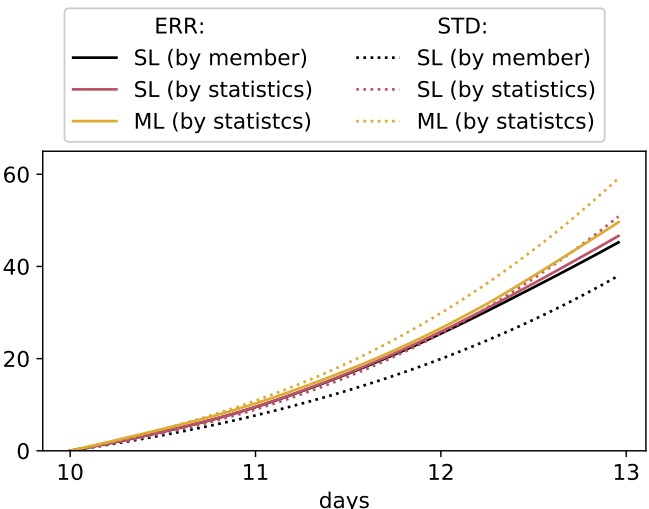

**Figure 11.** Calibration and spread of the drift trajectory forecasts.

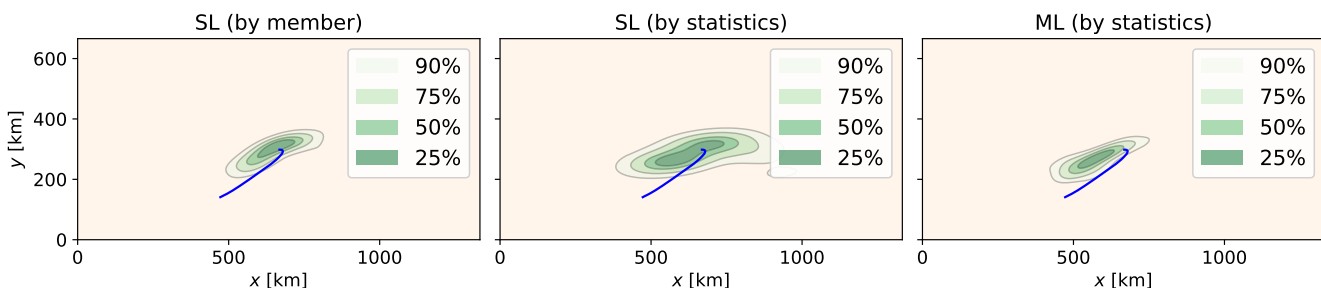

**Figure 12.** Levels of estimated kernel densities for the final position of drifter (c) from Figure 10. True trajectory is plotted in blue.

ensemble compared to the classical single-level approach with drifter per member first to the biased walk modelling and second to the state ensemble spread.

We have repeated the experiments with varying parameters for the biased walk and observed that for $N_d < 50$ drifters, results are not robust, while for $N_d > 50$ the improvements start to stagnate. Similarly, for drifter time steps up to approximately

$\Delta t = 5\,\mathrm{min}$ the results are qualitatively comparable.

From the final positions of the modelled drifters, the probabilities of their where-about can be calculated by kernel density estimation. Figure 12 shows the areas corresponding to the probability levels of 90, 75, 50 and 25% for drifter (c), along with its true trajectory in blue. The mode and the spread of the probability distributions differ between the methods, but we see as in Figure 10 that the single-level ensemble covers this particular drifter slightly better than the multi-level ensemble.

To investigate coverage properties further, we follow the idea of rank histograms, and we record how often one of the 100 simulated true drifters ends in each of the four probability areas. Table 2 shows the results for the same probability levels as


**Table 2.** Probabilities of the true drifters to be in the corresponding probability zone at the end of the forecast.

| Drifter | Probability levels | | | |
|---|---|---|---|---|
| Case | 25% | 50% | 75% | 90% |
| SL (by member) | 16 | 46 | 69 | 85 |
| SL (by statistics) | 26 | 55 | 74 | 90 |
| ML (by statistics) | 38 | 65 | 84 | 91 |

in Figure 12. The counting results suggest that the single-level with biased walk matches the right spread very well, while the classical single-level is slightly under-dispersive and the multi-level trajectories are slightly over-dispersive. This is in line with the results from Figure 11, where the error in the mean is the same for all three methods, but the spread in the ensembles differ

correspondingly at day 13 .

## 6 Concluding remarks

In this article, we have applied the MLEnKF successfully to a simplified ocean model. While MLDA has shown potential in a theoretical context and in proof-of-concept examples, we have focused towards applications with practical relevance in ocean forecasting. On the way, we have discussed various practical challenges that naturally arise when MLDA is implemented.

The goal of multi-level methods is to gain a computational speed-up with respect to a single-level method, while maintaining the same statistical error. First in Sect. 2, we have revised the MLEnKF by Chernov et al. (2021) for nested grids like in finite-volume methods. The practical considerations are discussed in detail in Sect. 3, and summary points are:

i) We re-used our existing GPU-accelerated simulator GPU Ocean, which is optimised for running single-level ensembles of high resolution, and we extended it by the integration of the MLEnKF. While a flexible choice of the resolution in the simulator enables to run ensembles on different levels, the devil was in the details, and an efficient choice of the model error became crucial for multi-level ensembles with coupled pairs.

As multi-level methods profit from lower computational costs on coarser levels, the performance scaling between the levels influences the speed-up directly. Even though the GPU framework in our example enables to run very fast simulations, it has suboptimal performance scaling for problems with very coarse grid resolutions. This reduced the potential of computational benefits through MLDA. Meanwhile, we were still interested in the upper bound of the computational speed-up, that may be obtainable for other software frameworks. We therefore considered and discussed the speed-up under assumptions of both theoretical and practical work. We obtained a theoretical speed-up up to a factor 2.

ii) For practical reasons, we fixed one multi-level ensemble throughout the full numerical experiments and used that one for the estimation of different quantities of interest at different times. For the multi-level ensemble configuration, we ran





a trial experiment with single-level data assimilation to record the variances per level at the final time and discussed two

quantities of interest that are purposeful for data assimilation. This choice again influenced the theoretical speed-up, but

we have seen that it does not change the assimilation and forecast quality in any relevant range.

iii) To assess the quality of multi-level data assimilation, we tested three different multi-level ensemble configurations with

same theoretic statistical quality as a given single-level ensemble. All three cases resulted in qualitatively the same mean

and spread compared to each other. Rank histograms from repeated short-span experiments indicated an appropriate fit

of the ensemble distribution.

Compared to the equivalent single-level experiment, the multi-level experiments gave similar error in the mean but

slightly larger spread for the partially observed momentum, but significantly lower error and spread for the unobserved

sea-surface elevation.

iv) We have introduced a drift model based on a so-called biased walk, that relies on the estimation of the mean and the

variance of the currents. Therefore, the posterior multi-level ensemble could be used for the relevant estimations and

we demonstrated how multi-level ensembles can be used for forecasting of drift trajectories. We compared the classical

single-level drifter modelling with one drifter per ensemble member to the biased walk approach, where we used both

the single-level and the multi-level ensemble for the estimations. The averaged results for 100 initial drifter positions

show that the error in the mean is qualitatively the same for all, but the spreads of the drifter ensemble differ.

With this article, we have laid the ground for a general discussion of the applicability of MLDA in ocean forecasting and we
have addressed the most prominent points to advance multi-level methods one step further towards practical relevance in data
assimilation.

*Code availability.* The source code used to produce the results presented in this paper is available under a GNU free and open source
license in order to enhance scientific exchange, see https://github.com/metno/gpuocean for the core and https://github.com/FlorianBeiser/
multilevelDA for the examples. Upon acceptance, we will upload the final version of the code to Zenodo and include the DOI here.

*Author contributions.* FB: Conceptualization, Methodology, Software, Validation, Writing - Original Draft, Visualization. HHH: Concep-
tualization, Methodology, Software, Validation, Writing - Original Draft, Supervision, Project administration, Funding acquisition. KOL:
Conceptualization, Methodology, Validation, Writing - Original Draft, Supervision. JE: Conceptualization, Methodology, Validation, Writ-
ing - Original Draft, Supervision.

*Competing interests.* The authors declare that they have no conflict of interest.





*Acknowledgements.* This work is part of the Havvarsel project supported by the Research Council of Norway under grant number 310515.





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
