# Peer review of "Multi-level data assimilation for simplified ocean models"

_Nonlinear Processes in Geophysics, 2023_

## Referee Comment (RC1)

**General comments:**

This study applied a multi-level Monte Carlo (MLMC)-based Ensemble Kalman filter (EnKF) to a shallow-water model, comparing it with a widely used single-level EnKF. The objective of this study is to demonstrate the applicability of the MLMC-based EnKF to practical systems, using the MLMC theory to replace a part of the original high-resolution ensemble forecasts with low-resolution forecasts, thereby reducing computational costs.

However, several significant issues are identified in the manuscript:

**1. Assimilation of Unavailable Observations:**

This paper assimilates unavailable observations of zonal and meridional momentum.

**2. Inconsistent Localization**

$P^f H^T$ localization is applied only to the first part of the Kalman gain, despite the presence of $HP^f H^T$ in the inversed part.

**3. Challenges in Practical Implementation**

Formulation based on the perturbed observation method results in an exceedingly large size matrix of the Kalman gain, posing challenges for practical system implementation and difficulties in parallel computation.

**4. Numerical Instabilities and Negative Eigenvalues**

Negative eigenvalues lead to numerical instabilities when applying eigenvalue decomposition to the inversed matrix in the Kalman gain. The likelihood of implementing the MLMC-based EnKF successfully is questionable.

**5. Unreasonable System Settings**

Because of the chosen system settings, such as the absence of stochastic external forcing, filter divergence is likely to occur as seen in decreasing the ensemble spread kept over the assimilation period. Therefore, the experimental period of 10 days was too short to conclude. In addition, there are other numerous issues such as inconsistency between the observation errors used for generating observations and the prescribed observation error covariance matrix.

**6. Lack of Statistical Tests**

The lack of statistical tests in the sensitivity experiments raises questions to significantly identify differences between the single- and multi-level Monte Carlo methods.

**7. Language and Presentation Issues**

The manuscript contains numerous instances of ambiguous expressions, typos, incorrect grammar, and a lack of definitions for words and mathematical symbols. It appears that co-authors may not have conducted a thorough review of the manuscript.

From the identified issues, I concluded that the paper falls short of meeting the standards expected for publication in international journals.

**Specific comments:**

[Abstract]

The authors have addressed practical challenges in implementing a multi-level Monte Carlo (MLMC)-based data assimilation method with a simplified shallow-water model to assimilate ocean zonal and meridional momentum. However, it should be noted that operational ocean data assimilation systems typically do not assimilate momentum or ocean current data (Balmaseda et al. 2015 and Martin et al. 2015). Consequently, it is recommended to modify the experimental settings and/or descriptions in this paper to better align with the current practices in operational ocean data assimilation.

The authors frequently employ expressions such as 'apply or incorporate observations to models' throughout the manuscript. However, it is important to note that data assimilation combines models and observations using dynamical systems and statistical theories, and therefore observations are assimilated in data assimilation systems. Therefore, it would be more appropriate to use terms such as 'assimilate observations in data assimilation systems' to accurately represent the underlying process. Please carefully revise the relevant descriptions throughout the manuscript to ensure consistency with the principles of data assimilation.

[Section 1]

First paragraph: It would be better to start by explaining what a single-level Monte Carlo (SLMC) method is and then move to the MLMC because the SLMC is a common data assimilation method. In addition, the authors should give simple and easy expressions to understand the MLMC. To me, at least the first paragraph in Section 1 is hard to read.

The first paragraph in Section 1 could benefit from a clearer structure by initially introducing the single-level Monte Carlo (SLMC) method, a widely used data assimilation approach, before transitioning to the multi-level Monte Carlo (MLMC) method. This approach would help readers build a more solid understanding. Furthermore, the authors are encouraged to provide simplified and easily understandable expressions for the MLMC method. This paragraph seems challenging to comprehend, and efforts should be made to enhance its readability.

L43-45: The optimal ensemble size would be infinity to remove spurious correlations caused by sampling errors (e.g., Kondo and Miyoshi 2016). It is crucial to explicitly mention what specific aspect is being optimized for in this context. The authors should

provide clarification on the optimization objective to enhance the understanding of the readers.

L44-45: The Kalman filter and ensemble Kalman filter are traditionally formulated under the assumption of no correlation between forecast and observation errors. The authors mentioned "correlations with observations" without specifying the details. It is essential to elaborate on the nature and specifics of these correlations with observations for a clearer understanding.

[Subsection 1.1]
L68-70: The authors' explanations seem inconsistent because variational methods are typically formulated to maximize the probabilistic density function (PDF) derived from backgrounds, observations, and these errors. To ensure clarity and coherence in the manuscript, it is recommended that the authors revisit and carefully align their explanations to maintain consistency with the fundamental principles of variational methods.

[Subsection 2.1]
Eq. (1): Since ¥delta x is used to represent ensemble perturbations and first-order terms in the Taylor expansion, it is suggested that model errors are represented by alternative variables such as q and ¥eta rather than ¥delta x.

L105-106: In addition to the "missing or unresolved physics in the model" as mentioned by the authors, it is essential to acknowledge that model errors can arise from various other factors. Therefore, the current description of model errors appears to be incomplete. It is recommended that the authors provide a more comprehensive discussion.

L107-108
The statement that 'the state x(t) will therefore represent grid cell averages of physical variables for all grid cells in the domain at time t' suggests an interpretation of x(t) as the average value over the entire domain. However, this may lead to confusion. It is recommended to clarify that x(t) represents the grid cell averages rather than a single average value for the entire domain at time t.

L110 and others: According to the formatting guidelines on the website (https://www.nonlinear-processes-in-geophysics.net/submission.html#math), it is

recommended to use 'Eq.' or 'Equation' before the formula number. Therefore, it is necessary to revise the manuscript to ensure consistency with the specified formatting rules.

Second paragraph: If regional high-resolution models have smaller dimensions than global coarse models, the computational costs of fine models would be smaller. Therefore, it is recommended to carefully reconsider and revise the explanations in the second paragraph.

L120: Please explain the definitions of "complete" and "incomplete" observations.

L120–121: The expression of "the true state … denoted xtrue" and "The model for the observation" are incorrectly phrased. Please clarify and rephrase these statements for accuracy.

L122-123: In Eq. (3), the bold font used for H suggests that H is a linear observation matrix. To avoid confusion, it is crucial to use precise expressions to distinguish between a matrix and a function.

L124-125: Please summarize the explanation of H in the previous sentence.

The last paragraph: Please consider removing the last paragraph because its content would be considered common knowledge among researchers in the data assimilation field.

L142-143: Please describe specifically, especially the latter half of the sentence.

L145: It should be clarified that ¥overbar{y} represents the ensemble mean in the observation space, not observations.

L159-160 and others: $HP^fH^T$ is not 'the cross covariance … predicted observations,' but a forecast error covariance matrix in the observation space ($P^f$: forecast error covariance matrix in the model space). Additionally, the authors' expression for $HP^fH^T+R$ is incorrect and requires appropriate revision.

L160 and all formulations: Considering operational ocean data assimilation systems with observation numbers exceeding $10^5$–$10^6$ per day, methods such as the reduced-order

Kalman filter (Fukumori et al. 1995) and local ensemble transform Kalman filter (LETKF; Hunt et al. 2007) have been proposed to reduce matrix sizes. The size of the Kalman gain as based on the perturbed observations is impractically large with dimensions proportional to the model size times observation size. Therefore, to enable the implementation of an MLMC-based EnKF with practical systems, I recommend exploring a formulation based on the LETKF rather than perturbed observation methods.

L163: If superscript a is used for analyses, it would be better to add superscript f for forecasts.

L172: Please define level 0.

L180: "merging … each". Did the authors average the values at west-, east-, north-, and south-side grids? Please clarify this procedure.

The authors frequently use characters without proper definitions. This is not permissible in scientific papers. It is imperative for the authors to meticulously review the manuscript and ensure that all symbols and characters are clearly defined just after the characters first appear.
- $N^l$ in Eq. (13).
- $\| \cdot \|_2$ in L217
- scalar w in L364
- subscripts t, x, y in Eq. (24)
- $\alpha^{sin}$ and $^{cos}$ in Eq. (26).
- $I_2$ in L457
- $c^L$ in L499
- $N_d$ in L607
- (a) in L618 and corresponding (b) and (c) etc.

L214: Please provide clarification on the statistical accuracy represented by $\tau^2$. Is it meant to denote RMSE (Root Mean Square Error) or other specific statistical values?

L217: Could you please provide a clearer definition for the L2-norm, specifically in terms of whether it involves summing up the squared state vector over time?

L219: While it is commonly known that larger ensemble sizes can reduce sampling errors caused by ensemble approximation, the authors described that the use of finer models can also lead to a reduction in sampling errors. It is crucial to provide a theoretical foundation for this description because it appears to deviate from the conventional understanding.

L238: Please clarify the meaning of "optimal".

Eqs. (19a) and (19b): Since $K^{ML}$ represents the Kalman gain for the finest system, there are inconsistencies in dimensions between the $x^0$ and analysis increments.

L265: Kalman gain is calculated from $P^f$ and $R$, and the description of "All these … Kalman gain matrix" is inconsistent.

[Subsection 3.1]
Figure 2: Please modify the expression to "relative A to B" in the caption of Fig. 2b. It is unclear what computation times are compared. In addition, at the resolution of $1024^2$, it seems that some computations might slightly exceed the theoretical limit. It is necessary to investigate and confirm this.

L304 and others: The use of expressions such as 'we see' and 'we observe' to describe the results introduces subjectivity. To maintain objectivity, it is recommended to remove or replace these expressions throughout the manuscript.

L307 and others: The use of 'very' is not suitable for scientific papers, as it introduces subjectivity without objectively quantifying the degree. It is suggested to replace or eliminate 'very' to enhance precision and objectivity.

[Subsection 3.1.2]
When referring to 'the mean,' please specify whether it denotes the ensemble mean, spatial mean, temporal mean, or another specific type of mean to avoid ambiguity.

Eqs. (20) and (21): If the equations are meant to indicate definitions, please use ¥equiv rather than ¥equal.

L314: If the authors conducted a twin experiment, it is recommended to use the terminology 'twin experiment' to accurately describe what kinds of experiments the

authors performed.

L345: Please provide clarification on the term 'different stochastic truths' to ensure a more precise understanding.

[subsection 3.1.3]
A more specific description of the derivation of Eq. (22) is necessary, as it is crucial for determining the ensemble size for each level. Providing additional details on the process would enhance the clarity and understanding of the manuscript.

[subsection 3.2]
L357–358 and others: The term 'performance' is used in the context of computation, whereas 'accuracy' is employed to describe the closeness of forecasts and analyses to true values. It is important to maintain consistency in the use of these terms throughout the manuscript to ensure clarity in communication.

L364: Please clarify the scalar w. Generally, the Gaspari and Cohn or Gaussian functions are used for the localization.

In the last paragraph, the authors implemented the localization by replacing $P^f H^T$ with $w$ ¥circle $P^f H^T$. However, $P^f H^T$ appears in two parts in the Kalman gain, and both should be localized. An alternative option is to implement K-localization by replacing K with $w$ ¥circle K. K-localization would offer a simpler and more consistent approach compared with the localization the authors applied.
Furthermore, the formulation of the ETKF and the implementation of the R-localization with the ETKF enable easy parallel computation. It would be challenging to implement parallel computation with the perturbed observation-based EnKF and the $P^f H^T$ localization. It seems that the authors did not apply parallel computation to EnKF, but parallel computation is an essential factor in constructing practical data assimilation systems.

[Subsection 3.3]
L375–379: The descriptions in L375-379 are not clear to me. Please provide a more careful and specific explanation for better comprehension.

When error covariances (i.e., off-diagonal elements) in matrices $P^f$ and R are substantial,

it is known that negative eigenvalues can be estimated when applying eigenvalue decomposition to $(HP^fH^T+R)^{-1}$. As a result, studies on observation error correlations (i.e., off-diagonal elements of R) often adopt reconditioning techniques to mitigate numerical instabilities (e.g., Weston et al. 2014; Tebeart et al. 2020).

In the case of the MLMC, where ensemble members are assumed to be independent, substantial forecast error covariances (i.e., off-diagonal elements in $P^f$) are not expected. Therefore, it is essential to investigate the causes of negative eigenvalues.

L406-407: Please use the terminology of "nature run" and specify the method to generate the observations (e.g. adding random noises).

L407: The authors mentioned 'use the same model to run the multi-level ensemble.' However, the system setting such as horizontal resolution and initial conditions should differ among multi-level forecasts. Please correct the description to accurately reflect the different system settings for each level.

[Subsection 4.1.1]
L420: The authors prescribed constant Coriolis parameter f (i.e., f-plane), and it is necessary to clarify the assumption the authors made. Furthermore, please specify the latitude corresponding to the prescribed f.

In addition to providing the grid size, it would be beneficial to include information on the horizontal resolution to understand the model configuration.

To ensure scientific reproducibility, it is essential to include detailed information on the resolution of both the coarser and coarsest models in the description of the model configuration.

The information provided on the method to generate the initial conditions of sea surface height is insufficient. Additionally, it is crucial to ensure that the initial conditions for the nature and forecast runs are different. If this condition is not met, the authors would be assuming an unrealistic situation in which the authors can obtain true values. Please provide more details on the method for generating the initial conditions of sea surface height and confirm the differences in the initial conditions between the nature and forecast run.

L430: It is essential to specify the method to generate model errors in this study. Typically, model errors are not substantial in twin experiments, as they result from imperfect factors in models as well as tangent linear and ensemble approximations in the KF and EnKF, respectively. If model errors are large enough to generate chaotic flows, the shallow-water model employed in this study is meant to not accurately represent spatiotemporal oceanic variations. Therefore, it is necessary to provide reasonable experimental settings, particularly regarding model errors.

The authors described that the shallow-water model behaves chaotically in Line 431. However, it is confirmed that the model without model errors and unique external forcing is less chaotic using the Lyapunov exponent. Although the current field appears to be chaotic at the initial stage of the experiment, attractors are likely to converge to a certain condition over time. Therefore, the experimental period of about 10 days would be too short to evaluate statistical indicators such as accuracy and ensemble spread. It is essential to conduct data assimilation experiments for a longer period over 1 month (might be over 1 year?) to show statistically significant results.

[Subsection 4.1.3]
Zonal and Meridional momentums (hu and hv, respectively) correspond to the vertical integration of zonal and meridional velocities in the 3D ocean and cannot be observed. This is inconsistent with the aim of this study to develop MLDA for practical systems. In addition, operational systems do not assimilate current data as shown in the review papers (Balmaseda et al. 2015; Martin et al. 2015). Therefore, it is necessary to provide consistent descriptions for observation variables to be assimilated in practical applications.

Standard deviations of observation errors prescribed as 0.1 m/s are not consistent with the prescribed observation error covariance matrix R=500I. It is necessary to provide a clear explanation for the reasons using inconsistent observation errors and observation error matrix in the experimental setting.

Is the chosen localization scale of 50 km optimal? It is required to investigate the optimal localization scale using simple systems to justify the selected scale.

Please specify "relaxation factor of 0.25". Does this indicate relaxation parameters in relaxation-to-prior perturbation and spread (RTPP and RTPS, respectively; Zhang et al. 2004; Whitaker et al. 2012)?

Fig. 5: No units in the y-axis. There are no blue and red lines.

$v^l$ and $V^l$ with g2 appear to represent the variance of variance of a variable. This meaning is unclear, and it is necessary to provide a clear explanation for better understanding.

Table 1: Please provide the details on how to decide the ensemble size for each level. Are the ensemble sizes in Table 1 are the optimal combination?

L487-488: Correlation with what? Similar to what?

L488-489: "as a coarser … information" is incorrect grammar.

L494: hu is dominating what?

L495-496: Please provide a clear explanation of "theoretical error" and "the same function".

Fig. 6: No units in the axis. "a 3h interval" should be "3 hours". There are no descriptions of experiments and lines.

Fig. 6: Since the ensemble spread of hu and hv gradually decreases and is not stable in the assimilation period, it is suggested that the experimental period is not sufficiently long. Extending the assimilation period further would lead to exceedingly small ensemble spread and filter divergence. Slight degradations can be found near the end of the assimilation period. Therefore, it is essential to conduct sufficiently long assimilation experiments and implement schemes to avoid filter divergence.
Furthermore, the accuracy of sea surface height is better in the ML experiments than the SL experiment, but this is not clear for hu and hv, especially hv. This is inconsistent because the sea surface height is directly linked with the current in the shallow water model. Therefore, it is crucial to investigate the cause of the inconsistency.

[subsection 4.3.1]
Eq. (28): The state variable x includes three variables with different units, and therefore Eq. (28) is inconsistent. Instead, it would be more appropriate to calculate RMSEs for

each variable.

L516-517: Not clear. Please provide a clear explanation.

L529–530: The authors described that the computation cost of the system is expensive. However, if high-performance and parallel computing are used, the cost of an LETKF-based shallow-water data assimilation system would not be expensive since the model is 2D.

L531-532: Even in the coarse system with 275 ensemble members, the localization would be necessary because of the presence of substantial sampling errors. Therefore, it is essential to demonstrate that the ensemble size of 275 is sufficient to reduce the spurious correlations.

[subsection 4.3.2]
Line 542: Please indicate a specific section of the discussion of the difference between MLDA and MLMC.

It is necessary to describe the meaning of $V^l/v^l$ to clarify the discussion in subsection 4.3.2.

[subsection 4.3.3]
The experiments conducted in this study should be summarized in the Method Section.

Please provide a clear explanation of "truth's function value" in Line 562, "true observation value" in Line 563-564, and "the indicator function" in Line 567.

[subsection 5.1]
It is more natural to use analysis velocity estimated from Eq. (30) than Eq. (31) because Eq. (31) results in the analysis velocity with positive biases. Although the authors seem to provide the reasons not to adopt Eq. (30) in the third paragraph, these reasons are not entirely clear. Therefore, it is necessary to describe the reasons more carefully and specifically.

Please provide a clear explanation of "prediction of the true drifter" in L623-624 and "ensemble calibration" in Line 627.

Equation (32) represents error variance, not error. Please provide appropriate expression.

Fig. 11: No label and unit in the y-axis.

The forecast accuracy of the trajectory is better in the SL experiment than in the ML experiment (Fig. 11), whereas that of the current shows no substantial differences between the SL and ML experiments (Fig. 6). Therefore, it is necessary to investigate the cause of the better trajectory accuracy in the SL experiment.

[Section 6]
To demonstrate that the accuracy in the single-level experiment is almost the same as in the multi-level experiments, it is essential to apply statistical tests to investigate the significant differences among the sensitivity experiments.

**Technical corrections:**

It is imperative to convey information with scientific precision, providing clear and objective descriptions in international journals. However, upon reviewing this paper, I have identified numerous instances of ambiguous expressions, typos, incorrect grammar, and a lack of definitions for words and mathematical symbols as listed. It is crucial for the authors to meticulously refine the manuscript. I express concern that the current state of the paper might not meet the standards for the review process. It seems plausible that co-authors may not have thoroughly checked the manuscript. It is strongly recommended to consider engaging the services of professional English editing to ensure the manuscript meets the requisite standards for publication.

L16: Please specify "By harnessing … within the ensemble".

L18: Please specify "the different levels". Which levels did the authors indicate accuracy, resolution, or vertical levels?

L22: Do "partial observations" indicate sparse observations? If so, please correct it with "sparse" throughout the manuscript.

L22: What does "the system" stand for? Please describe specifically.

L22-23: "work" should be "works", and "has" should be "have".

L23: Please describe specifically "potential gains".

L24: "distinguish" should be "distinguishes".

L28-29: "generalized circulation model" should be "general circulation models" or "Ocean general circulation models".

L30-31: "input fields" should be "ocean current data", and "a deterministic … at best" should be "deterministic or small ensemble forecasts".

L31: "one has suggested" should be "Roed (2012) suggested".

L31-33: "larger" is not appropriate because this sentence does not include a target to

compare.

L32-33: It appears to duplicate the meaning of "to investigate … ocean currents" and "quantify … trajectories".

L35: Remove "apply and".

L36: "running ensembles" should be "conducting ensemble forecasts" or "conducting ensemble simulations".

L41: Remove "run".

L42: Please specify "run time scaling".

L44: "others" should be "other".

L47: "equivalent … ensembles" should be "traditional single-level data assimilation methods".

L51-52: Duplicate the meaning of "For uncertainty quantification" and "for the estimation of expected values".

L53: Remove "by".

L53: "telescopic sum" might be "telescoping sum".

L55: Please specify "budget".

L57: "case" should be "cases".

L61: "The nature of the methods" should be "The kinds of ensemble data assimilation methods". Remove "commonly split into".

L62: "filters" should be "filter".

L64-65: Incorrect grammar.

L66: "ensemble member" should be "particle".

L81: Please specify what "a particular challenging test case" is and why "challenging" is.

L82: Insert "will demonstrate to" between "work" and "bring".

L84: Please specify "robust".

L85: Please specify "Among others".

L85: "work with" should be "make"

L86: "work" should be "works".

L86: If "it" indicates "the assumption", "it" should be "they".

L88: Please specify "opposing jets".

L89: "state estimation quality" should be "accuracy", and remove "of corresponding theoretical error".

L102: "representation of the same system" should be "model".

L105: Remove "term".

L118: "isolated model time step" would be "analysis time steps".

L131: No definition of "N" in x_e.

L136: "ensemble anomalies" should be "ensemble perturbations".

L136: "covariance" should be "error covariance matrix".

L141: "comma" should be "and".

L166: "Monte Carlo variability" should be "sampling error".

L194: "model perturbations" should be "model errors".

L222: Remove "it takes".

L237: "Given … $\{C\_l\}^L\_{l=0}$" should have the form of "A, B, and C".

L240 "in" should be "to".

L280: "standard deviation" should be "spread".

L280: "are" might be "is".

L280: Please specify "robust statement".

L282: Please specify "Assessment scores"

L303: Please insert "size" between "large" and "problem".

L310 and others: "best" should be "the best".

L319: It is not good to cite something that comes later.

L323-324: Incorrect grammar.

L326: "foe" should be "for".

L340: "N" should be "N^l".

L359: "analyzed for" should be "applied to".

L381: Please check the spell of "assemblance".

L392: "that" should be "than".

L406: Replace "spread" with "ensemble spread" and remove "around the mean".

L418-419: Please add ", respectively" at the end of the sentence.

Axis in Fig. 4: "Velocity" should be "Speed".

Line 453: "3d" should be "3 days". This is true for the similar expressions throughout the manuscripts.

L483: "is" should be "are".

L528: Please specify "the curves".

L533: "but" should be "except for".

L543: "that" should be "the".

Please specify the meaning of "directly" in Line 581.

Incorrect grammar in L635-637 and L638-639.

Please specify "the devil" in Line 660.

Line 672: "assimilation" should be "analysis".

**References:**

Balmaseda, M.A., Hernandez, F., Storto, A., Palmer, M.D., Alves, O., Shi, L., Smith, G.C., Toyoda, T., Valdivieso, M., Barnier, B., Behringer, D., Boyer, T., Chang, Y.S., Chepurin, G.A., Ferry, N., Forget, G., Fujii, Y., Good, S., Guinehut, S., Haines, K., Ishikawa, Y., Keeley, S., Köhl, A., Lee, T., Martin, M.J., Masina, S., Masuda, S., Meyssignac, B., Mogensen, K., Parent, L., Peterson, K.A., Tang, Y.M., Yin, Y., Vernieres, G., Wang, X., Waters, J., Wedd, R., Wang, O., Xue, Y., Chevallier, M., Lemieux, J.F., Dupont, F., Kuragano, T., Kamachi, M., Awaji, T., Caltabiano, A., Wilmer-Becker, K., Gaillard, F., 2015. The ocean reanalyses intercomparison project (ORA-IP). J. Oper. Oceanogr. 8, s80–s97.

Martin, M.J., Balmaseda, M., Bertino, L., Brasseur, P., Brassington, G., Cummings, J., Fujii, Y., Lea, D.J., Lellouche, J.M., Mogensen, K., Oke, P.R., Smith, G.C., Testut, C.E., Waagbø, G.A., Waters, J., Weaver, A.T., 2015. Status and future of data assimilation in operational oceanography. J. Oper. Oceanogr. 8, s28–s48.

Kondo, K., Miyoshi, T., 2016. Impact of removing covariance localization in an ensemble Kalman Filter: Experiments with 10 240 members using an intermediate AGCM. Mon. Weather Rev. 144, 4849–4865.

Fukumori, I., Malanotte-Rizzoli, P., 1995. An approximate Kaiman filter for ocean data assimilation: An example with an idealized Gulf Stream model. J. Geophys. Res. 100, 6777.

Hunt, B.R., Kostelich, E.J., Szunyogh, I., 2007. Efficient data assimilation for spatiotemporal chaos: A local ensemble transform Kalman filter. Phys. D 230, 112–126.

Weston, P.P., Bell, W., Eyre, J.R., 2014. Accounting for correlated error in the assimilation of high-resolution sounder data. Q. J. R. Meteorol. Soc. 140, 2420–2429.

Tabeart, J.M., Dance, S.L., Lawless, A.S., Nichols, N.K., Waller, J.A., 2020. Improving the condition number of estimated covariance matrices. Tellus, Ser. A Dyn. Meteorol. Oceanogr. 72, 1–19.

Zhang, F., Snyder, C., Sun, J., 2004. Impacts of initial estimate and observation availability on convective-scale data assimilation with an ensemble Kalman filter. Mon. Weather Rev. 132, 1238–1253.

Whitaker, J.S., Hamill, T.M., 2012. Evaluating methods to account for system errors in ensemble data assimilation. Mon. Weather Rev. 140, 3078–3089.

---

## Referee Comment (RC2)

**General Comments:**

This paper proposes the innovative application of Multi-Level Monte Carlo (MLMC) and Multi-Level Data Assimilation (MLDA) techniques within the realm of simplified ocean model based on the shallow-water equations, aiming to enhance computational efficiency while maintaining or improving forecast accuracy. The authors effectively highlight the theoretical foundations of MLMC and MLDA, compare these methods to traditional single-level approaches, and discuss the integration of GPU-accelerated frameworks to address the computational demands of high-resolution simulations. Despite its contributions, the manuscript suffers from structural and clarity issues that need addressing. Consequently, my recommendation is for publication following major revisions.

**Major Comments:**

1. The paper lacks necessary explanations for several terms. For instance, there is hardly any description of MLMC, MLDA, and their single-level counterparts. The differences between MLDA and MLMC are not clearly stated. Discussions on GPUs and CPUs appear suddenly without prior introduction or context. Additionally, the structure of sections and subsections is complex, making some parts difficult to read, especially the introduction section. In the equations, the use of superscripts and subscripts is prevalent, but their application seems cluttered in places, indicating room for improvement.

2. One of the primary objectives of this paper is the proposal of a new method to improve computational efficiency. However, there is insufficient comparison of computation time and computational costs between this new method and traditional methods. It would be beneficial to identify which aspects of the traditional methods incur significant computational costs and how much time the proposed method takes in comparison. Comparing computational costs and error scores between single-level and multi-level approaches could verify whether this method is truly effective and practical. Furthermore, discussing potential issues and differences when applying this method to actual ocean models, in addition to idealized experiments, would be beneficial.

**Detailed Comments:**

1. Section 1: What is MLMC? Please provide a brief explanation.

2. L16: 'By harnessing the cost-effectiveness of low-fidelity simulations within the ensemble'—please elaborate on this statement with specific details.

3. Section 1: Can you explain the benefits of performing MLDA, compared to traditional methods?

4. L26: Since 'search-and-rescue (SAR)' hardly appears in the text, defining the abbreviation 'SAR' may not be necessary unless it is used more extensively.

5. Section 1 is divided into three parts, making the content quite difficult to understand. Rather than dividing it, reconsidering and organizing the content in a more orderly manner into a single section would likely make it easier for readers to comprehend.

6. Section 1.1: A brief mention of the relationship between data assimilation and MLMC could improve the connection between the first and second paragraphs.

7. L74: What is MLEnKF, and how does it differ from the traditional EnKF?

8. L84: What does 'robust data assimilation' mean?

9. L87: What is a GPU? What is meant by a GPU-accelerated framework, and how do data assimilation and MLMC relate to GPUs?

10. L88: What are 'sparse observations'?

11. L117: 'In sequential data assimilation, the state is ...'— this statement seems specific to 3D data assimilation and not applicable to 4D methods like smoothers. The same applies to line 152.

12. L123: You are using **H** (in bold font), indicating a linear observation operator. Is it possible to use *H* for nonlinear observations in practice?

13. L120: What is 'The model for the observation'?

14. L130: Please explain the meaning of 'single-level'.

15. L166: What ensemble update method do you use? Perturbed observation method? Square root filter? Or something else?

16. Section 2.3: Before entering this section, could you briefly state which parts of the computation cost are problematic in normal Single-level Monte Carlo EnKF, and how using Multi-level Monte Carlo aims to avoid these issues?

17. L242: Considering the frequent use of superscript throughout, it might be better to avoid the expression $K^{(ML)}$.

18. L282: 'Assessment scores may also be used to evaluate the quality of the ensemble-based representation, and these tasks require various kinds of functions.' I'm having trouble understanding this sentence.

19. L290: In the long sentences in Section 3.1, the most crucial statement appears to be 'we choose the ensemble size tailored for the Kalman gain estimation in the MLEnKF,' yet no concrete method is discussed. Please provide a detailed explanation.

20. Section 3.1.1: I don't understand the necessity of this section. What is the relation between Multi-level data assimilation and GPUs or CPUs? How does this section's content relate to the equations used in Multi-level data assimilation? Could you explain which specific parts of the equations in this paper contribute to the difference in computational costs?

21. The structure of Section 3.1 (and Section 3) is very confusing. What role does Section 3.1.2 play within it?

22. Would it be better to add the content of Section 3.1.3 at the end of Section 3.1?

23. L359: Please explain "level-local formulations."

24. Section 3.2: Do you use vertical localization? If not, why is it unnecessary?

25. Section 3.3: Please discuss further the impact of negative eigenvalues in real applications.

26. Section 3: Is inflation used?

27. L459: What is "a relaxation factor"? Why is inflation or relaxation necessary?

28. L465: Rather than solely relying on visual comparisons, would converting the truth shown for 10d in Fig3 to the appropriate grid size and comparing them with the results in Fig4, including calculating scores, provide a more objective evaluation?

29. L487-489: I'm having trouble understanding this passage.

30. Figure 6: In hu and hv, the results of single-level experiments and multi-level experiments are similar. Can you provide theoretical insights into what this means and what it implies?

31. Figure 6: Why does the difference between ERR and STD become significant after around day 8 for all variables?

32. Figure 6: The scores are not stable during this experimental period. Is the experimental period too short?

33. L532: "It is reasonable to do so for the coarsest level of case (A) with 275 members." Really? Why do you think that?

34. L537: "Once the dynamics becomes more turbulent around day 6 to 7," Why is that? Is there a reason for this setting? What's the reasoning behind this setting? Sorry if it was mentioned somewhere and I missed it.

35. L542: "As discussed before, MLDA differs from MLMC" - which part are you referring to? Also, a brief explanation here again might make it easier for readers to understand.

36. L544: Could you briefly explain what the value 'V^l/v^l t' represents and its significance in the context of your study?

37. Figure 8: Why does the relative variance of the variable eta in the experiment with the darkest blue line gradually decrease after day 10?

38. L564: What does "For the true observation value y" mean?

39. Figure 9: By plotting the results of conventional methods over the MLEnKF results, it might be possible to compare the two methods and further highlight the effectiveness of MLEnKF.

40. L624: What does 'outlier' mean?

41. Figure 10: The results of the three methods are very similar, making it hard to see the differences in this figure. You have Figure 11, so is Figure 10 necessary? As you mention in L626, especially in this experiment with a short spin-up and experimental period, it heavily depends on the initial values. So, I think the results of Figure 10 highly depends on initial ensembles.

42. L631: "We notice that the calibration curves for all methods are very close, but the spread using multi-level methods is a little bit bigger." This expression is ambiguous.

43. Section 5: One of the main reasons for using a new method in this study was thought to be improving computational efficiency. How about comparing computation time and computational cost in this setting and experiment? Discussing whether this method is truly practical by comparing computational load and error scores between SL and ML could be helpful.

44. L654: "we have discussed various practical challenges that naturally arise when MLDA is implemented." Could you please explain this part in detail?

45. L660: "the devil was in the details" Could you please explain this part in detail?

46. Section 6: Please describe the potential issues when applying your method to actual models and observations.

---

## Author Comment (AC1)

Dear reviewer and NPG community,

Thank you for reviewing our paper. To foster the possibility of having constructive discussions about our work[1], we first reply and suggest action items to the seven "Significant issues" identified by the reviewer as listed under "General comments". In a subsequent post, we would like to take the opportunity to provide point-by-point answers to all your comments, including the "Specific comments" and "Technical corrections".

Before getting to the listed issues, we would like to point out that we clearly stated the *scope of the paper*, including four clearly marked research questions in L26 – L49, which we return to and answer at the end of the paper on L658 – L685. Furthermore, the *contribution of the paper* is described on L79 – L91 in Section 1.2 "Contribution and Outline". As far as we can see, none of the reviewer's general comments seem to discuss or acknowledge the scope of our research, the research questions we aim to answer, or the novel contribution we claim to bring to the scientific community. We are disappointed, and in fact a bit puzzled, that these elements are not discussed in this review.

Rather than reviewing our paper, the reviewer criticizes the established building blocks of our work. Hence, as will be clear from our reply to each of the raised "significant issues", we point out that several of the reviewer comments are outside the scope of our paper on multi-level Monte Carlo and assimilation of sparse ocean data.

Additionally, we point to NPG's aims & scope[2]. The journal is "devoted to breaking the deadlocks often faced by standard approaches in Earth and space sciences", and to apply innovative concepts and methodologies to address the complexity in geoscience systems. We believe that our application of a multi-level ensemble Kalman filter applied to a simplified ocean model fits very well into this. In light of this, we would like to invite the editor to also participate in this discussion.

Yours sincerely,
Florian Beiser
On behalf of all the authors
* * *
[1]as encouraged by NPG, see `https://www.nonlinear-processes-in-geophysics.net/peer_review/interactive_review_process.html`

[2]`https://www.nonlinear-processes-in-geophysics.net/about/aims_and_scope.html`

**Replies to General comments**

**1. Assimilation of Unavailable Observations**

**Reviewer:** *This paper assimilates unavailable observations of zonal and meridional momentum.*

**Reply:** In our work, we use an identical twin experiment to assess the applicability of the multi-level ensemble Kalman filter (MLEnKF). Contrary to most of the literature on multi-level Monte Carlo, we apply the method to a shallow water equation (SWE) model which is more relevant to oceanographic applications than previous examples that are of a theoretical flavor, see the original presentations Hoel et al. (2016); Chernov et al. (2021). The setup uses sparse observations of a subset of the physical model variables, i.e. the momentum at 50 locations in the domain. This design is in agreement with the experiment's purpose.

In the introduction, L24 – 34, we explain how our work is motivated by search-and-rescue (SAR) operations at sea, and how ensembles of simplified ocean models can be used complementary to available operational circulation forecasts. The in-situ observations mentioned in this regard, can be made by local buoys or drifters (such as Rabault et al. (2022)) released from the SAR vessel. To assimilate such observations into SWE models, they must be mapped from observational space (velocity) to model space (momentum), where the associated observation error accounts for the representation error in this mapping. As a step towards these types of applications, we have in our work shown the applicability of MLEnKF for a relevant model through an identical twin experiment.

**Action item:** We will make the purpose of our experimental setup clearer in the text. We will also improve the description of our motivation in the introduction to better show the relevance of our experiments.

**2. Inconsistent Localization**

**Reviewer:** $P^f H^T$ *localization is applied only to the first part of the Kalman gain, despite the presence of* $H P^f H^T$ *in the inversed part.*

**Reply:** We use Kalman gain localisation which has been commonly done in data assimilation, see e.g. Chen and Oliver (2010) or Chapter 10.4 in Evensen et al. (2022). In comparison with other localization approaches, this type is natural for our particular situation with sparse data and a large model space (see also our reply to Comment 3).

**Action item:** We will improve text description and add references on this topic, which is established in the literature.

**3. Challenges in Practical Implementation**

**Reviewer:** *Formulation based on the perturbed observation method results in an exceedingly large size matrix of the Kalman gain, posing challenges for practical system implementation and difficulties in parallel computation.*

**Reply:** A characteristic of our case is sparse observational data, meaning that the matrices required in the Kalman gain are not problematically large. Within this regime, the computationally expensive part is in fact the forward calculation of the differential equations of the state (and particularly so at the fine-resolution level). This is why we employ multi-level methods combined with GPU-accelerated models.

It is not the intention that our implementation of the MLEnKF will replace the operational data assimilation systems for integrating all available data. Instead, we study tools and methods capable of assimilating subsets of observational data within ensembles, in this case at multiple levels. We argue that ensembles of (efficient) simplified models has a role to play by complementing the large operational model systems for specialized applications (such as sparse data in SAR operations). See our introduction, L29 – 34. We also write in our abstract L4 – 5, *By applying a multi-level ensemble Kalman filter for assimilating **sparse observations** of ocean currents....*

Due to this, the reviewer's comment falls outside the scope of our paper. We also point out that the contribution of our work is to make *a new step on the path of making multi-level data assimilation feasible for real-world oceanographic applications* (Abstract, L7 – 8), and we do not hide the fact that there are more steps to be taken in the future for advancing the applicability of MLEnKF further.

**Action item:** We will clarify both in the introduction and in the Kalman updating expressions that our work is within the regime of sparse observational data.

**4. Numerical Instabilities and Negative Eigenvalues**

**Reviewer:** *Negative eigenvalues lead to numerical instabilities when applying eigenvalue decomposition to the inversed matrix in the Kalman gain. The likelihood of implementing the MLMC–based EnKF successfully is questionable.*

**Reply:** Other researchers, before our paper, have recognized the challenges of estimating covariances within the framework of multi level Monte Carlo (MLMC) methods, with no guarantees for getting positive definite covariance matrices (all eigenvalues larger than 0), see e.g., Maurais et al. (2023) and Shivanand (2023) and the references therein. Building on the established understanding, we honestly discuss this challenge in Section 3.3 (L375 – 376):

*While the empirical measure associated with a classical Monte-Carlo*

> *estimator is positive in the sense that all Dirac contributions have
> a positive sign, this is not the case for multi-level estimators where
> the differences between the levels can introduce negative values.*

When other researchers in future work aims to further develop a MLMC-based
data assimilation scheme that depends on eigenvalue decomposition, we believe
that our discussion in Section 3.3, along with that of others, will be a helpful
resource for them to find a way to understand this challenge and formulate
solutions.

As we discuss in Section 3.3 (L391 – 392), the issue with negative eigen-
values is infrequent (less than 1:10 000 times of the assimilated observations).
For sparse observations in particular, we show that the known problem can be
handled in a pragmatic way, and that potential errors introduced in the process
are negligible.

**Action item:** We will add references to the appropriate literature early in
Section 3.3, to help explain that this is a known challenge.

**5.  Unreasonable System Settings**

**Reviewer:** *Because of the chosen system settings, such as the absence of
stochastic external forcing, filter divergence is likely to occur as seen in de-
creasing the ensemble spread kept over the assimilation period. Therefore, the
experimental period of 10 days was too short to conclude. In addition, there
are other numerous issues such as inconsistency between the observation errors
used for generating observations and the prescribed observation error covariance
matrix.*

**Reply:** While we agree that our experimental setup is not described well enough
(as illustrated by some of the reviewer's "Specific comments"), we do not agree
with the statement that we cannot make any conclusions.

The aim of the experimental design was to generate ensembles of arbitrary
sizes with turbulent and (initially) chaotic behavior, in which drift trajectory
forecasts made through pure Monte Carlo experiments would have too high
uncertainties to be helpful, but where successful data assimilation improves the
forecasts significantly. Since the mentioned decrease in the ensemble spread
occurs in both the single- and multi-level experiment (Fig. 6 of our paper),
we do not see how this affects the scientific contribution of our paper on the
applicability of the MLEnKF. Moreover, since our work is motivated by SAR
operations at sea, assimilation and forecast periods of seven and three days,
respectively, are already in the upper time-range relevant for this application.

This experiment was first outlined in a paper presenting a GPU-accelerated
implicit equal-weight particle filter (Holm et al., 2020), which we refer to in
our paper. Here, we should have pointed out for the reader that more details
concerning the experiment could be found in that paper, along with a pure

Monte Carlo experiment and data assimilation experiments with too sparse and/or inaccurate observations to give improved forecasts.

On the specifically mentioned inconsistency (related to L457), our intention was to use a standard deviation around 0.1 m/s in the current velocities. To map this value to the observation error covariance matrix, which acts on the momentum, we multiply by the equilibrium depth (230 m) and square the result, which gives us $\mathbf{R} = 529\ \mathbf{I}$. Since this value is arbitrarily chosen, we fixed $\mathbf{R} = 500\ \mathbf{I}$ (both in the observation error and the observation error matrix). We see that our description of this was inaccurate, and it will be clarified when revising the manuscript.

**Action item:** We will add more details about the experimental design and setup, possibly by also including an appendix. We will also carefully go through the specific comments from the reviewer related to this comment.

**6. Lack of Statistical Tests**

**Reviewer:** *The lack of statistical tests in the sensitivity experiments raises questions to significantly identify differences between the single- and multi-level Monte Carlo methods.*

**Reply:** The statistical properties of the MLEnKF has been analysed in theoretical settings by Chernov et al. (2021). Our work aims at exploring the applicability of MLEnKF, focusing on the practical challenges of applying the method to more applied geophysical problems than what has been done before. We keep the single-level EnKF as a reference, and the main result of the paper is and will still be related to the available speed-up of introducing multiple levels. The fact that MLEnKF works at all for this type of model is a novel contribution to both the multi-level and data-assimilation communities.

But going beyond this main result of speed-up, we have, before submitting our paper, conducted tens of replicate runs related to the statistics of the outputs and the sensitivity of the MLMC settings. The results from all these tests have had qualitatively similar results as the reference experiment that we used in the paper. We will however summarize the key findings of these replicate runs in the revised paper.

**Action item:** We will add a discussion around our replicate studies in the simulation study.

**7. Language and Presentation Issues**

**Reviewer:** *The manuscript contains numerous instances of ambiguous expressions, typos, incorrect grammar, and a lack of definitions for words and mathematical symbols. It appears that co-authors may not have conducted a thorough review of the manuscript.'*

**Reply:** We thank the reviewer for pointing out specific grammatical mistakes and suggestions for clarifying the notation. We are prepared to carefully modify and improve the text and notation of the paper.

**Action item:** We will carefully go through the text of the paper and also let a native English person read through and provide comments to us.

**References**

Chen, Y. and Oliver, D. S. (2010). Cross-covariances and localization for enkf in multiphase flow data assimilation. *Computational Geosciences*, 14:579–601.

Chernov, A., Hoel, H., Law, K., Nobile, F., and Tempone, R. (2021). Multi-level ensemble Kalman filtering for spatio-temporal processes. *Numerische Mathematik*, 147(1):71–125.

Evensen, G., Vossepoel, F. C., and van Leeuwen, P. J. (2022). *Data assimilation fundamentals: A unified formulation of the state and parameter estimation problem*. Springer Nature.

Hoel, H., Law, K., and Tempone, R. (2016). Multilevel ensemble kalman filtering. *SIAM Journal on Numerical Analysis*, 54(3):1813–1839.

Holm, H. H., Sætra, M. L., and Van Leeuwen, P. J. (2020). Massively parallel implicit equal-weights particle filter for ocean drift trajectory forecasting. *Journal of Computational Physics: X*, 6:100053.

Maurais, A., Alsup, T., Peherstorfer, B., and Marzouk, Y. (2023). Multi-fidelity covariance estimation in the log-euclidean geometry. *arXiv preprint arXiv:2301.13749*.

Rabault, J., Nose, T., Hope, G., Müller, M., Breivik, Ø., Voermans, J., Hole, L. R., Bohlinger, P., Waseda, T., Kodaira, T., Katsuno, T., Johnson, M., Sutherland, G., Johansson, M., Christensen, K. H., Garbo, A., Jensen, A., Gundersen, O., Marchenko, A., and Babanin, A. (2022). OpenMetBuoy-v2021: An easy-to-build, affordable, customizable, open-source instrument for oceanographic measurements of drift and waves in sea ice and the open ocean. *Geosciences*, 12(3).

Shivanand, S. K. (2023). Covariance estimation using h-statistics in monte carlo and multilevel monte carlo methods. *arXiv preprint arXiv:2311.01336*.